# Plant–necrotroph co-transcriptome networks illuminate a metabolic battlefield

**Wei Zhang[1,2], Jason A Corwin[3], Daniel Harrison Copeland[2], Julie Feusier[2], Robert Eshbaugh[2], David E Cook[1], Suzi Atwell[2], Daniel J Kliebenstein[2,4]***

[1]Department of Plant Pathology, Kansas State University, Manhattan, United States; [2]Department of Plant Sciences, University of California, Davis, Davis, United States; [3]Department of Ecology and Evolution Biology, University of Colorado, Boulder, United States; [4]DynaMo Center of Excellence, University of Copenhagen, Frederiksberg, Denmark

**Abstract** A central goal of studying host-pathogen interaction is to understand how host and pathogen manipulate each other to promote their own fitness in a pathosystem. Co-transcriptomic approaches can simultaneously analyze dual transcriptomes during infection and provide a systematic map of the cross-kingdom communication between two species. Here we used the Arabidopsis-*B. cinerea* pathosystem to test how plant host and fungal pathogen interact at the transcriptomic level. We assessed the impact of genetic diversity in pathogen and host by utilization of a collection of 96 isolates infection on Arabidopsis wild-type and two mutants with jasmonate or salicylic acid compromised immunities. We identified ten *B. cinerea* gene co-expression networks (GCNs) that encode known or novel virulence mechanisms. Construction of a dual interaction network by combining four host- and ten pathogen-GCNs revealed potential connections between the fungal and plant GCNs. These co-transcriptome data shed lights on the potential mechanisms underlying host-pathogen interaction.

DOI: https://doi.org/10.7554/eLife.44279.001

**\*For correspondence:**
kliebenstein@ucdavis.edu

## Introduction

How a host and pathogen manipulate each other within a pathosystem to facilitate their own fitness remains a long-standing question. The difference between the pathogen's ability to infect and the host's ability to resist generates the resulting disease symptomology. This interaction forces host-pathogen dynamics to shape the genomes of the two species via adaptive responses to each other (*Dangl and Jones, 2001*; *Bergelson et al., 2001*; *Benton, 2009*; *Kanzaki et al., 2012*; *Karasov et al., 2014*). Plants have evolved a sophisticated set of constitutive and inducible immune responses to cope with constant selective pressures from antagonistic microbes (*Jones and Dangl, 2006*). Reciprocally, plant pathogens have also evolved a variety of different invasion and virulence strategies to disarm or circumvent plant defense strategies (*Glazebrook, 2005*; *Toruño et al., 2016*). This has resulted in complex relations between plant hosts and fungal pathogens for survival and fitness.

The plant innate immune system includes several functional layers with overlapping functions to detect and defend against phytopathogens. This multi-layer immune system can be categorized as a signal monitor system to detect invasion, local and systemic signal transduction components to elicit and coordinate responses, and defensive response proteins and metabolites focused on combatting the invading pathogen (*Tsuda and Katagiri, 2010*; *Corwin and Kliebenstein, 2017*). These functional layers, as well as the components within them, are highly interconnected and tightly regulated

**eLife digest** Infections are complex interactions between two organisms. When a disease-causing microbe and a potential host engage, molecules continuously flow in both directions. This creates an inter-connected loop of messages and counter-messages, attacks, counter-attacks and resistance. This communication determines the final winner and the outcome of the disease. Yet it is technically difficult to measure it from both organisms at the same time, mostly because it is often impossible to tell whether a given molecule came from the microbe or the host. As such, little is known about how most infections play out at the molecular level.

Now, rather than looking directly at the communication molecules, Zhang et al. have measured the active genes in samples of a plant infected with a fungus. While a molecule released by the plant may be indistinguishable from one from the fungus, the genes needed to make those molecules will be different in each species. The experiments involved two species where databases of gene sequences already exist: *Arabidopsis thaliana*, a plant often used in laboratory studies, and a fungus known as *Botrytis cinerea*, which infects many plants.

Zhang et al. showed that the interactions between the two organisms are diverse and, rather than single genes, they largely involve sets of genes that are all switched on together as so-called gene co-expression networks (or GCNs for short). Ten of these networks encoded mechanisms that allow the fungus to attack plant hosts. Further analysis identified potential connections between networks of genes in the plant and fungus. These connections may reveal some of the targets of the fungus's toxins or counter mechanisms that plants can use to attempt to defend themselves.

These findings show that it is possible to listen to the molecular communication between two organisms during an infection. In the future, a similar approach may make it possible to ask if a host plant communicates with all of its possible disease-causing microbes with a few distinct pathways, or if instead, hosts have the flexibility to uniquely communicate with each microbe in a different way.

DOI: https://doi.org/10.7554/eLife.44279.002

by the host plant to respond appropriately to various phytopathogens (*Couto and Zipfel, 2016*; *Tang et al., 2017*). For instance, Arabidopsis utilizes a complex signaling network to regulate the production of indole-derived secondary metabolites, such as camalexin and indole glucosinolates, that contribute to resistance against pathogens (*Kliebenstein et al., 2005*; *Clay et al., 2009*; *Bednarek et al., 2009*; *Frerigmann et al., 2016*; *Xu et al., 2016*; *Mine et al., 2018*). This layered immune system provides pathogens with numerous targets in the plant immune system that the pathogen can utilize, evade or attack. Most biotrophic pathogens, evolved from commensal microbes, attempt to dismantle the plant immune system by injecting effector proteins into host cells or the inter-cellular space (*Dangl and Jones, 2001*; *Büttner and He, 2009*; *Stergiopoulos and de Wit, 2009*). For example, the biotrophic bacterial pathogen *Pseudomonas syringae* can utilize the jasmonic acid (JA) signaling pathway through the production of a JA-mimic, coronatine, to enhance its fitness (*Mittal and Davis, 1995*; *Brooks et al., 2005*; *Cui et al., 2018*). Alternatively, necrotrophic pathogens, which often evolved from environmental saprophytic microbes, can utilize toxic secondary metabolites, small secreted proteins, and small RNAs to aggressively attack host defenses while also defending against host-derived toxins (*Choquer et al., 2007*; *Arbelet et al., 2010*; *Mengiste, 2012*; *Weiberg et al., 2013*; *Kubicek et al., 2014*; *Macheleidt et al., 2016*). In addition, pathogens can directly resist downstream defenses as is done by *B. cinerea*, where it has an ATP-binding cassette (ABC) transporter BcatrB that provides resistance by exporting camalexin from the pathogen cell (*Stefanato et al., 2009*). This high level of interactivity between the immune system and pathogen virulence mechanisms generates the final level of disease severity. However, a functional description of this combative cross-kingdom communication between a plant host and necrotrophic pathogen remains elusive.

Co-transcriptomic approaches whereby the host and pathogen transcriptomes are simultaneously analyzed provide the ability to systematically map the cross-kingdom communication between plant hosts and their pathogens, both for individual genes and gene co-expression network (GCN) levels (*Stuart et al., 2003*; *Musungu et al., 2016*; *Zhang et al., 2017*; *Lanver et al., 2018*; *McClure et al., 2018*). Recent advances have enabled the measurement of pathogen *in planta* transcriptome. For

example, *in planta* measurements of the pathogens' transcriptome within the biotrophic Arabidopsis-*Pseudomonas syringae* pathosystem has enabled the investigation of early effects on Arabidopsis host immunity and the consequent effects on bacterial growth (*Nobori et al., 2018*). This enabled the identification of a bacterial iron acquisition pathway that is suppressed by multiple plant immune pathways (*Nobori et al., 2018*). This shows the potential for new hypothesis to be generated by a co-transcriptome approach (*Swierzy et al., 2017*; *Westermann et al., 2017*; *Lee et al., 2018*).

The Arabidopsis-*B. cinerea* pathosystem is well suited for exploring plant-pathogen interaction to understand host defenses and necrotrophic virulence in ecological and agricultural settings. *B. cinerea* is a necrotrophic generalist pathogen that attacks a broad range of diverse plant hosts, including dicots, gymnosperms, and even bryophytes (*Williamson et al., 2007*). This necrotrophic pathogen is endemic throughout the world and can cause severe pre- and post-harvest losses in many crops. A high level of standing natural genetic variation within *B. cinerea* population is hypothesized to facilitate the extreme host range of *B. cinerea*. This genetic variation affects nearly all known *B. cinerea* virulence strategies, including penetration and establishment, evading detection, and combatting/coping with plant immune responses (*Atwell et al., 2015*; *Walker et al., 2015*; *Corwin et al., 2016b*). For example, a key virulence mechanism is the secretion of phytotoxic secondary metabolites, including the sesquiterpene botrydial (BOT) and the polyketide botcinic acid (BOA) that trigger plant chlorosis and host cell collapse (*Deighton et al., 2001*; *Colmenares et al., 2002*; *Wang et al., 2009*; *Rossi et al., 2011*; *Ascari et al., 2013*; *Porquier et al., 2016*). These metabolites are linked to virulence, but some pathogenic field isolates fail to produce either compounds pointing to additional pathogenic strategies. The combination of a high level of genetic diversity and extensive recombination means that a population of *B. cinerea* is a mixed collection of virulence strategies that can be used to interrogate by the co-transcriptome.

In the present study, the Arabidopsis-*B. cinerea* pathosystem is used to test how the transcriptomes of the two species interact during infection and assess how natural genetic variation in the pathogen impacts disease development. Isolates were inoculated on Arabidopsis Col-0 wild-type (WT) in conjunction with immune-deficient hormone mutants *coi1-1* (jasmonate defense signaling) and *npr1-1* (salicylic acid defense signaling). A collection of 96 isolates of *B. cinerea* was used for infection, which harbor a wide scope of natural genetic variation within the species (*Atwell et al., 2015*; *Corwin et al., 2016a*; *Zhang et al., 2016*; *Corwin et al., 2016a*; *Zhang et al., 2017*; *Soltis et al., 2018*; *Fordyce et al., 2018*). From individual infected leaves, both Arabidopsis and *B. cinerea* transcripts at 16 hr post-infection (HPI) were simultaneously measured. Arabidopsis transcripts was analyzed previously to identify four host-derived GCNs that are sensitive to natural genetic variation in *B. cinerea* (*Zhang et al., 2017*). In present analysis, ten fungal pathogen-derived GCNs were identfied, which encode either known or novel virulence mechanisms within the species. Some of these *B. cinerea* GCNs responsible for BOT production, exocytosis regulation and copper transport are highly linked with the host's defense phytohormone pathways. By combining the plant host- and pathogen-GCNs into a single network, a dual-transcriptomic network was constructed to identify potential interactions between the components of plant host innate immune system and fungal pathogen virulence. These connections highlight potential targets for fungal pathogen phytotoxins and prevailing counter-responses from plant host. Collectively, co-transcriptomic analysis shed lights on the potential mechanisms underlying how the host and pathogen combat each other during infection and illustrate the continued need for advancements of *in planta* analysis of dual-species interaction.

## Results

### Genetic variation in pathogen and hosts influence *B. cinerea* transcriptome

To investigate how genetic variation within a pathogen differentially interacts with plant host immunity at the transcriptomic level, we profiled the *in planta* transcriptomes of 96 *B. cinerea* isolates infection across three host genotypes, the Arabidopsis accession Col-0 WT and two immune-signaling mutants *coi1-1* and *npr1-1* that are respectively compromised in JA or salicylic acid (SA) driven immunity. This previously described collection of 96 isolates represents a broad geographical distribution and contains considerable natural genetic variation that affects a diversity of virulence

strategies within *B. cinerea* (*Denby et al., 2004*; *Rowe and Kliebenstein, 2007*; *Atwell et al., 2015*; *Corwin et al., 2016b*; *Zhang et al., 2016*). Four independent biological replicates across two separate experiments per isolate/genotype pair were harvested at 16HPI for transcriptome analysis. A total of 1152 independent RNA samples were generated for library preparation and sequenced on Illumina HiSeq platform (NCBI accession number SRP149815). These libraries were previously used to study Arabidopsis transcriptional responses to natural genetic variation in *B. cinerea* (*Zhang et al., 2017*). Mapping the dual-transcriptome reads against the *B. cinerea* reference genome (B05.10), we identified 9284 predicted gene models with a minimum of either 30 gene counts in one isolate or 300 gene counts across 96 isolates. The total of identified genes corresponds to ~ 79% of the 11,701 predicted encoding genes in B05.10 reference genome (*Van Kan et al., 2017*). The two different thresholds allowed the identification of pathogen transcripts that express only in a specific isolate.

Measuring the abundance of individual pathogen transcripts in relation to the host transcripts can be used as a molecular method to estimate fungal biomass (*Blanco-Ulate et al., 2014*). Given this, we hypothesized that the fraction of total reads that map to *B. cinerea* might be a biologically relevant indicator of pathogen virulence (*Figure 1—source data 1*). Comparing *B. cinerea* transcript abundance at 16HPI to lesion development at 72HPI revealed a significant partial correlation in the WT Col-0 ($R^2$ = 0.1101, p-value=0.0016, *Figure 1*). In contrast to WT, the early transcriptomic activities of most *B. cinerea* isolates were more vigorous in the two Arabidopsis mutants, resulting in a significant curvilinear relationship between total fraction of *B. cinerea* reads and final lesion area (p-value=3.914e-07, p-value=0.0001, respectively, *Figure 1*). Interestingly, the total reads fraction was better correlated with final lesion area in *coi1-1* ($R^2$ = 0.2562) than either WT ($R^2$ = 0.1101) or *npr1-1* ($R^2$ = 0.161). This suggests that early transcriptomic activity from the pathogen can be a partial indicator of pathogen virulence, but also depends on the respective resistance from the plant host.

Plant defense phytohormone networks, like SA and JA, help shape the immune responses of a plant host while also shape the virulence gene expression within bacterial pathogens, such as *Pseudomonas syringae* (*Nobori et al., 2018*). To test how variation in host SA/JA-singaling influences the fungal pathogen transcriptome, we applied a generalized linear model linked with negative-binomial function (nbGLM) to each *B. cinerea* transcript across the experiment. This analysis allowed us to estimate the relative broad-sense heritability ($H^2$) of genetic variation from the pathogen, plant host, or their interaction contributing to each transcript (*Figure 2—source data 1–3*). Of the 9284 detectable *B. cinerea* transcripts, 8603 and 5244 transcripts were significantly influenced by genetic variation in pathogen and host, respectively (74% and 45% of predicted *B. cinerea* gene models, respectively) (*Figure 2A*, *Figure 2—source data 3* and *4*). While this result shows that the plant phytohormone pathways influence *B. cinerea* gene expression, the variation in host defense responses (average $H^2_{Host}$ = 0.010) has far less influence on *B. cinerea* gene expression than that of the pathogens' own natural genetic variation (average $H^2_{Isolate}$ = 0.152). The host defense hormones also affected *B. cinerea* gene expression in a genotype-by-genotype dependent manner on 4541 genes (39% of *B. cinerea* predicted gene models, average $H^2_{Isolate \times Host}$ = 0.116) (*Figure 2B–2I*). Illustrating this potential for host x pathogen interactions on pathogen gene expression are the two genes encoding the well-studied polygalacturonase 1 (*Bcpg1*) and oxaloacetate acetyl hydrolase. The two virulence associated genes showed dramatic expression variation across 96 isolates in different host backgrounds (*Figure 3*, *Figure 3—figure supplement 1*, and *Figure 2—source data 1*). Extending this to 500 genes showing the strongest host x pathogen effect showed that there is a wide range of patterns that differs in the host *coi1-1* or *npr1-1* background with diverse pathogen strain specific patterns (*Figure 4*). One potential complication of this analysis is for sequence variation between the reference B05.10 genome and the diverse strains to create artificially low expression estimates. However, very few genes showed consistently low expression within a strain and instead when a gene showed no expression in one host genotype, it was expressed in a different host genotype (*Figure 3* and *Supplementary file 1*). This conditionality argues against a sequencing error as the sequence has not altered. The genes that did show a loss of expression across all host genotypes within a strain (i.e. BOT and BOA genes) were frequently linked to whole gene deletions that abolished their expression (*Soltis et al., 2019*). Thus, while there are likely some sequence variation associated expression errors, they are not a dominant signature in the data. Thus, within the Arabidopsis/*B. cinerea* pathosystem the pathogens transcriptional responses are influenced by a blend of the pathogens' natural variation and its interaction with the host, while there is less evidence for the host's

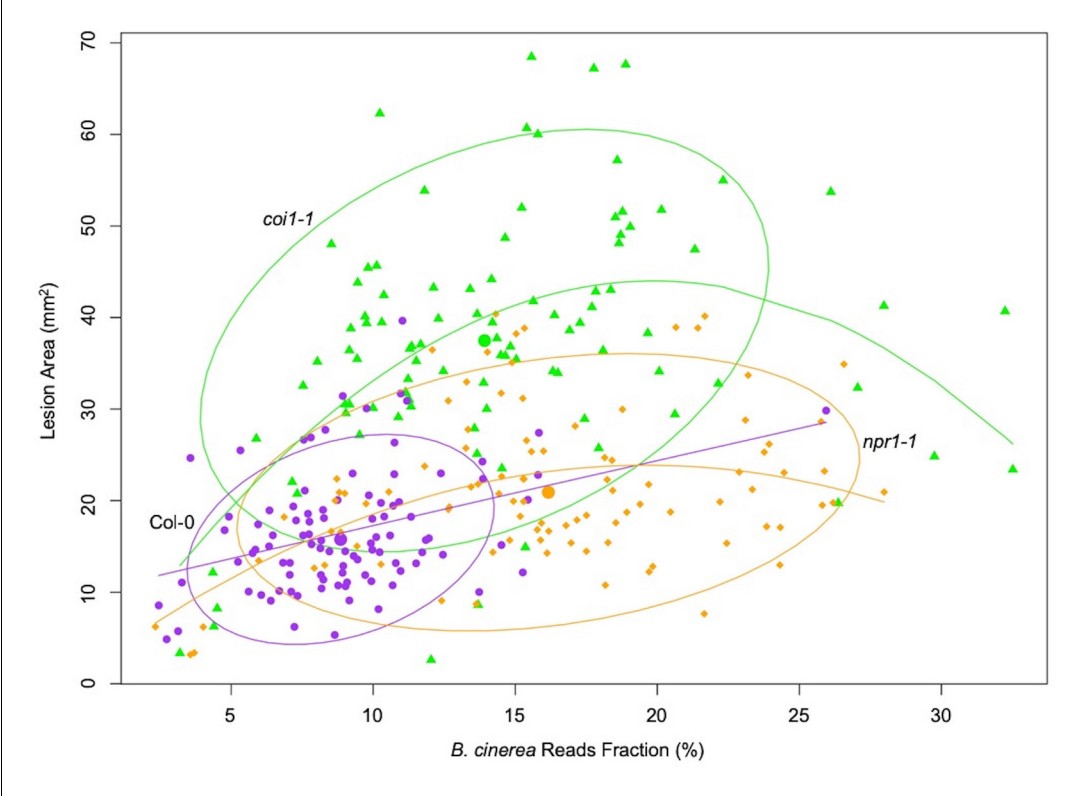

**Figure 1.** Correlation between earlier estimated *B. cinerea* biomass and later lesion area. Model-corrected lesion area means were estimated using the linear model on the six replicates data from three Arabidopsis genotypes at 72 hr post-infection with 96 *B. cinerea* isolates. Estimated biomass of *B. cinerea* was calculated using the linear model-corrected fraction of *B. cinerea* mapped reads against total mapped reads to Arabidopsis and *B. cinerea* reference genomes. RNA-Seq analysis was conducted at 16 hr post-infection for each pathosystem. Three Arabidopsis genotypes are wild-type Col-0 (purple dot), jasmonate insensitive mutant *coi1-1* (green triangle), and salicylic acid insensitive mutant *npr1-1* (orange diamond). The 90% confidence ellipse intervals are plotted for each Arabidopsis genotype for references. Quadratic regression lines are: Col-0: $y = -0.00059 \times^2 + 0.729\,x + 10.037$, p=0.0016, adjusted $R^2$ = 0.1101; *coi1-1*: $y = -0.117 \times^2 + 4.44\,x - 0.1585$, p=3.914e-07, adjusted $R^2$ = 0.2562; *npr1-1*: $y = -0.0579 \times^2 + 2.26\,x + 1.673$, p=0.0001, adjusted $R^2$ = 0.161.

DOI: https://doi.org/10.7554/eLife.44279.003

The following source data is available for figure 1:

**Source data 1.** Model-corrected means of estimated *B. cinerea* biomass.

DOI: https://doi.org/10.7554/eLife.44279.004

defense responses to unilaterally affect *B. cinerea*. Future work will hopefully assess how this extends to other host-pathogen systems.

## Identification of virulence factors among the early *B. cinerea* transcripts

This data set also allows us to test for specific *B. cinerea* transcripts whose early expression is associated with later lesion development. These genes can serve as potential biomarkers of overall pathogen virulence and may elucidate the functional mechanisms driving early virulence in the interaction. To find individual pathogen transcripts link with lesion development, we conducted a genome-wide false discovery rate-corrected Spearman's rank correlation analysis between 72HPI lesion area and individual *B. cinerea* transcripts accumulation at 16HPI. We identified 2521 genes (22% of *B. cinerea* predicted gene models) with significant positive correlations and 114 genes (1% of *B. cinerea* predicted gene models) with significant negative correlations to lesion area across three Arabidopsis genotypes, respectively (p-value<0.01, *Figure 3—source data 1*). The top 20 positively correlated *B. cinerea* genes contained all seven genes involved in BOT biosynthesis (*Deighton et al., 2001*; *Colmenares et al., 2002*; *Wang et al., 2009*; *Rossi et al., 2011*; *Ascari et al., 2013*; *Porquier et al., 2016*). In addition to phytotoxins, more than 30 genes of the top 100 lesion-

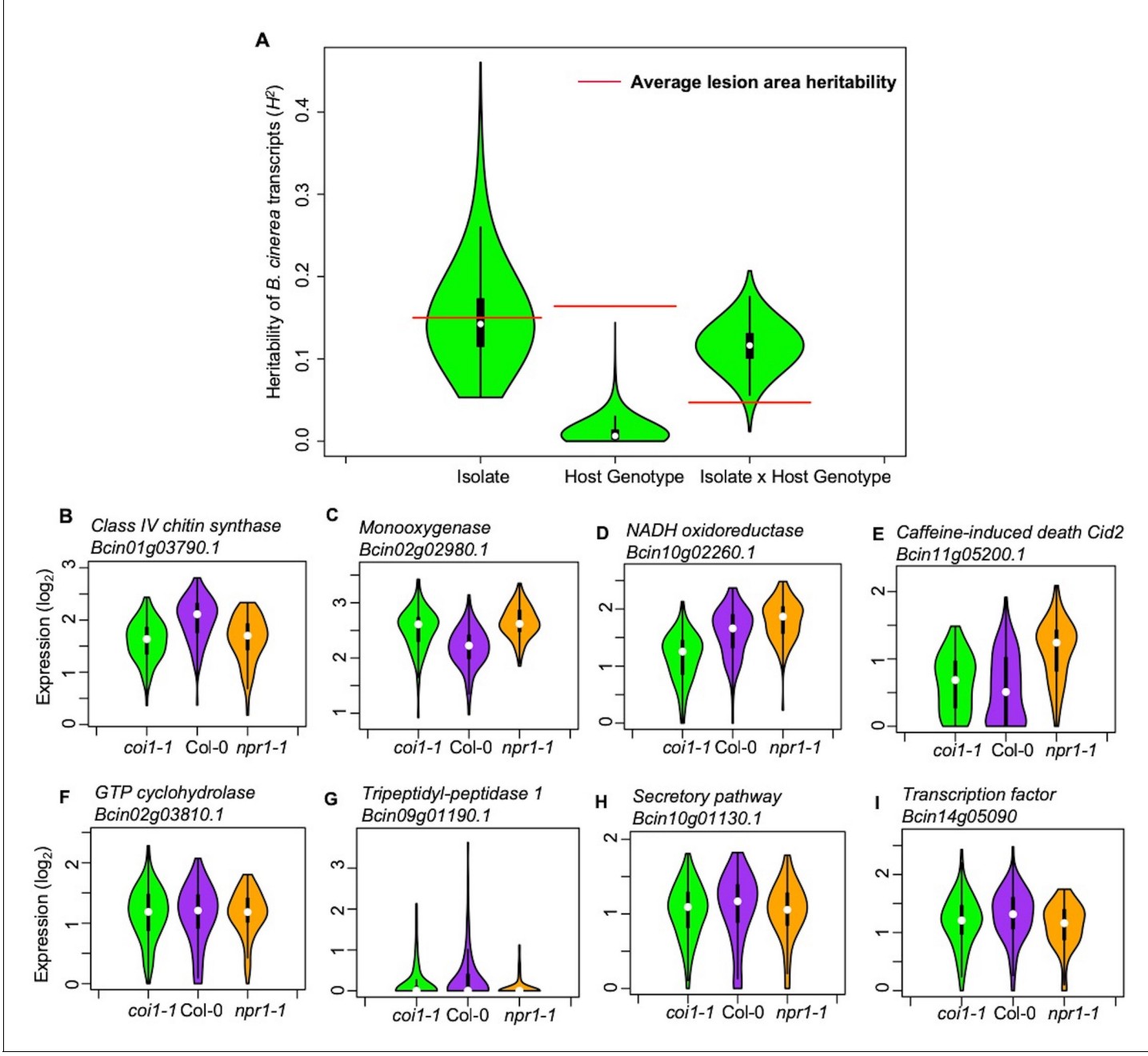

**Figure 2.** Transcriptomic responses of *B. cinerea* on Arabidopsis are controlled by genetic variation in pathogen population, host genotypes, and their interaction. (A) Distribution of broad-sense heritability ($H^2$) of *B. cinerea* transcripts contributed by genetic variation in the *B. cinerea*, Arabidopsis genotypes, and the interaction between pathogen and host. Violin plots illustrating the distribution of $H^2$ for transcripts from 96 *B. cinerea* isolates infecting on Arabidopsis genotypes. Heritability is partitioned across the different sources, 96 pathogen genotypes = 'Isolate', plant genotypes Col-0, *coi1-1* and *npr1-1* plant genotypes = 'Host', and the corresponding interaction. The transcriptomic analysis was conducted by sequencing mRNA extracted from *B. cinerea* infected Arabidopsis leaves at 16 hr post-infection. Red lines indicate the average broad-sense heritability values of lesion area caused by isolates, Arabidopsis genotypes, and their interaction. (B) to (E) Expression profiles of *B. cinerea* transcripts significantly influenced by host genotypes. The model-corrected means ($\log_2$) for *B. cinerea* transcript were used for plotting. The Arabidopsis genotypes, wild-type Col-0 (purple), jasmonate insensitive mutant *coi1-1* (green), and salicylic acid mutant *npr1-1* (orange), are shown on the x axis. *B. cinerea* transcripts are: (B) *Bcin01g03790.1*, class IV chitin synthase; (C) *Bcin02g02980.1*, Monooxygenase; (D) *Bcin10g02260.1*, NADH oxidoreductase; (E) *Bcin11g05200.1*, caffeine-induced death Cid2; (F) to (I) Expression profiles of *B. cinerea* transcripts significantly influenced by the interaction between pathogen and host genotypes. (F) *Bcin02g03810.1*, GTP cyclohydrolase; (G) *Bcin09g01190.1*, Tripeptidyl-peptidase 1; (H) *Bcin10g01130.1*, in secretory pathway; (I) *Bcin14g05090.1*, a transcription factor.

DOI: https://doi.org/10.7554/eLife.44279.005

*Figure 2 continued on next page*

*Figure 2 continued*

The following source data is available for figure 2:

**Source data 1.** Model-corrected means of *B. cinerea* transcripts.
DOI: https://doi.org/10.7554/eLife.44279.006
**Source data 2.** Standard errors of *B. cinerea* transcripts.
DOI: https://doi.org/10.7554/eLife.44279.007
**Source data 3.** GLM deviance tables and broad-sense heritability of *B. cinerea* transcripts.
DOI: https://doi.org/10.7554/eLife.44279.008
**Source data 4.** Top 100 heritability of *B. cinerea* transcripts.
DOI: https://doi.org/10.7554/eLife.44279.009

correlated genes encode plant cell wall degrading enzymes, that is glucosyl hydrolases, carbohydrate esterases, cellobiose dehydrogenases and *Bcpg1* (*Figure 3* and *Figure 3—source data 1*) (*Gerbi et al., 1996*; *Zamocky et al., 2006*; *Cantarel et al., 2009*; *Van Vu et al., 2012*; *Igarashi et al., 2014*; *Morgenstern et al., 2014*; *Blanco-Ulate et al., 2014*; *Tan et al., 2015*; *Courtade et al., 2016*; *Nelson et al., 2017*; *Pérez-Izquierdo et al., 2017*). Additionally, 10 of the top 100 lesion-correlated genes were annotated as putative peptidase activities, which are critical for fungal virulence (*Movahedi et al., 1991*; *Poussereau et al., 2001a*; *Poussereau et al., 2001b*; *ten Have et al., 2004*; *ten Have et al., 2010*). A final classical virulence gene in the top 100 gene list is *Bcoah* (*Bcin12g01020*) encoding oxaloacetate acetyl hydrolase, which is a key enzyme in oxalic acid biosynthesis that positively contributes to virulence (*Figure 3—figure supplement 1* and *Figure 3—source data 1*) (*Greenberg et al., 1994*; *Williamson et al., 2007*; *Walz et al., 2008*; *Schumacher et al., 2012*; *Schumacher et al., 2015*; *Tayal et al., 2017*). In addition, this method identified 37 of the top 100 lesion-correlated genes with no gene ontology (GO) terms, which likely represent unknown virulence mechanisms (*Figure 3—source data 1*). Thus, this approach readily creates new hypothesis about known and novel pathogen virulence functions.

## *In Planta* virulence Gene Co-expression Networks (GCNs) in *B. cinerea*

To develop a systemic view of fungal pathogen *in planta* gene expression, we used a co-expression approach to identify *B. cinerea* networks that associated with growth and virulence *in planta*. Using solely *B. cinerea* transcriptome at 16HPI from Arabidopsis Col-0 WT infected leaves, we calculated Spearman's rank correlations of gene counts across all *B. cinerea* isolates, filtered gene pairs with correlation greater than 0.8. We then used the filtered gene pairs as input to construct GCNs. We identified ten distinct GCNs containing more than five *B. cinerea* genes (*Figure 5*, *Supplementary file 1*, *Figure 5—figure supplement 1* and *Figure 5—source data 1*). The largest GCN with 242 genes contains members responsible for phospholipid synthesis, eiosome function, and membrane-associated stress signaling pathways (*Figure 5*-Vesicle/virulence). The biological function of this GCN suggests its role in fungal membrane- and vesicle-localized processes, which are normally involved with general hyphae growth, fungal cell wall deposition, and exudation of fungal toxins to the intercellular space (*Figure 5—source data 1*). The second largest network contains 128 genes that were entirely associated with translation and protein synthesis (*Figure 5*-TSL/growth and *Figure 5—source data 1*). Of the smaller GCNs identified (5–20 genes), five networks were identified with genes distributed across *B. cinerea* 16 chromosomes, suggesting that these GCNs arise from coordinated trans-regulation (*Figure 5*-Trans-networks and *Figure 5—figure supplement 1D, F and H–J*). These networks are associated with diverse array of virulence functions, including the regulation of exocytosis, copper transport, the production of peptidases and isoprenoid precursors (IPP), and polyketide secretion.

In contrast to the whole-genome distributed GCNs, three of the smaller GCNs were predominantly comprised of genes tandemly clustered within a single chromosome with no or a few genes on other chromosomes (*Figure 5*-BOA, -Cyclic Peptide, -BOT, *Figure 5—figure supplement 1C, E and G*). A functional analysis showed that all of the genes within these networks encoded known or putative biosynthetic enzymes for specialized metabolic pathways. For example, seven genes responsible for BOT biosynthesis cluster on chromosome 12 and form a small GCN with a Zn(II) 2Cys6 transcription factor that is specific to the pathway (*Figure 6A and B*, *Figure 5—figure*

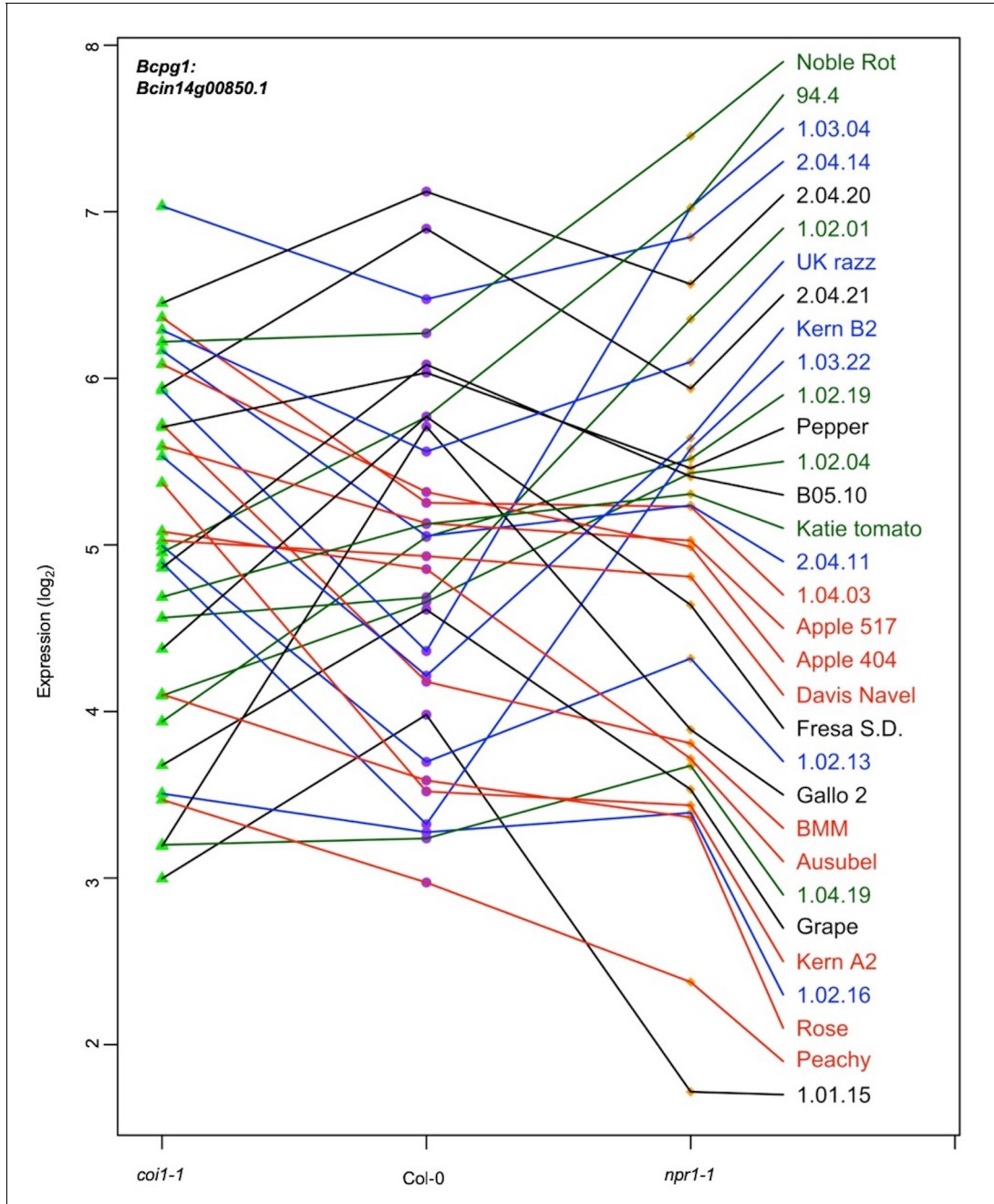

**Figure 3.** Expression profiles of an endopolygalacturonase gene *Bcpg1* from diverse *B. cinerea* isolates across Arabidopsis genotypes. Rank plot shows the relationship of *Bcpg1* expression from 32 diverse *B. cinerea* isolates (right) across three Arabidopsis genotypes (x axis). Three Arabidopsis genotypes are wild-type Col-0 (purple dot), jasmonate insensitive mutant *coi1-1* (green triangle), and salicylic acid mutant *npr1-1* (orange diamond). The model-corrected means (log$_2$) for the transcript of *Bcpg1* (*Bcin14g00850.1*) encoding an endopolygalacturonase gene are utilized for plotting. The transcript expression levels from the same isolate across three Arabidopsis genotypes are connected with a colored line. The names of 32 isolates are represented with the same colored lines as induced *Bcpg1* expression levels. Black lines indicate the expression levels of *Bcpg1* are higher in *coi1-1* and *npr1-1* than in Col-0. Red lines indicate the higher expression levels of *Bcpg1* in *coi1-1* but lower in *npr1-1*. Blue lines indicate the highest expression levels of *Bcpg1* are in Col-0. Dark green lines indicate the higher expression levels of *Bcpg1* in *npr1-1* but lower in *coi1-1*.
DOI: https://doi.org/10.7554/eLife.44279.010

The following source data and figure supplement are available for figure 3:

**Source data 1.** Spearman's rank correlation between lesion area and *B. cinerea* transcripts abundance.
DOI: https://doi.org/10.7554/eLife.44279.012

**Figure supplement 1.** Expression profiles of an oxaloacetate hydrolase gene *Bcoah* from diverse *B. cinerea* isolates across Arabidopsis genotypes.
*Figure 3 continued on next page*

*Figure 3 continued*

DOI: https://doi.org/10.7554/eLife.44279.011

supplement 1G and *Figure 5—source data 1*) (*Siewers et al., 2005*; *Pinedo et al., 2008*; *Urlacher and Girhard, 2012*; *Moraga et al., 2016*). Similarly, all 13 genes involved in BOA biosynthesis cluster in Chromosome one and form a highly connected GCN (*Figure 5*-BOA, *Figure 6E*, *Figure 5—figure supplement 1C* and *Figure 5—source data 1*) (*Dalmais et al., 2011*; *Porquier et al., 2019*). In addition to previously characterized secondary metabolic pathways, we identified an uncharacterized set of ten genes that cluster on Chromosome 1 (*Figure 5*-Cyclic Peptide, *Figure 6F*, *Figure 5—figure supplement 1E* and *Figure 5—source data 1*). These genes share considerable homology with enzymes related to cyclic peptide biosynthesis and may represent a novel secondary metabolic pathway in *B. cinerea* (*Figure 5—source data 1*). The expression of these pathways *in planta* was extremely variable among the isolates and included some apparent natural knockouts in the expression of the entire biosynthetic pathway (*Figure 6G* and *Figure 2—source data 1*). Isolate 94.4 was the sole genotype lacking the entire BOT pathway, while 19 isolates and 24 isolates did not transcribe respectively the BOA and the putative cyclic peptide pathways (*Figure 6E–6G* and *Figure 2—source data 1*). We decomposed the expression of these pathways into expression vectors, referred to as eigengenes, using a principle component analysis and used a linear mixed model to test for a relationship between early expression of secondary metabolic pathways and later lesion area. This showed a significant relationship between the expression of BOT and BOA pathways and lesion area measured at 72HPI (*Supplementary file 2*). In contrast, the putative cyclic peptide pathway was only associated with lesion development in a BOT-dependent manner, suggesting that it may have a synergism to BOT (*Supplementary file 2*). Thus, *in planta* analysis of the fungal transcriptome can identify known and novel potential virulence mechanisms and associate them with the resulting virulence.

## Covariation of fungal virulence networks under differing plant immune responses

The *B. cinerea* GCNs measured within Arabidopsis WT provide a reference to investigate how phytohormone-signaling in host innate immunity may shape the pathogen's transcriptional responses during infection. Comparing the *B. cinerea* GCN membership and structure across the three Arabidopsis genotypes (WT, *coi1-1*, and *npr1-1*) showed that the core membership within networks was largely maintained but the specific linkages within and between GCNs were often variable (*Figure 7*, *Supplementary file 1*, *Figure 7—figure supplements 1*, *2* and *3*, and *Figure 5—source data 1*). For example, the two largest *B. cinerea* GCNs in WT developed multiple co-expression connections during infection in the JA-compromised *coi1-1* host (*Figure 7* and *Supplementary file 1*). In contrast, some GCNs have a highly robust structure across three host genotypes, including three GCNs associated with BOT, BOA and cyclic peptide production, and GCNs associated with exocytosis regulation, copper transport, and peptidase activity (*Supplementary file 1*, *Figure 7—figure supplements 1*, *2* and *3*, and *Figure 5—source data 1*). In addition, we also identified additional small GCNs that demonstrated host specificity in *coi1-1* (*Figure 7—figure supplement 2* and *Figure 5—source data 1*). In particular, there were four small GCNs that are associated with plant cell wall degradation, siderophores, glycolysis, ROS, and S-adenosylmethionine biosynthesis (*Figure 7—figure supplement 2*). Thus, the coordinated transcriptional responses of *B. cinerea* GCNs are at least partially dependent on variation in the host immune response.

Host immunities showed different impacts on expression profiles of genes condensed in individual *B. cinerea* GCNs (*Figure 7—figure supplement 4*). Compared with WT, expression profiles of genes within the largest membrane/vesicle virulence GCN were elevated in the SA- and JA-compromised Arabidopsis mutants on average (*Figure 7—figure supplement 4A*). Fungal genes associated with copper transport and polyketide production were upregulated under SA-compromised host immunity (*Figure 7—figure supplement 4F and J*). Whereas, members of GCNs responsible for plant cell wall degradation and siderophore biosynthesis were upregulated under JA-compromised host immunity (*Figure 7—figure supplement 4K and L*). Finally, GCNs associated with BOT and exocytosis regulation showed robust gene expression profiles across all three Arabidopsis

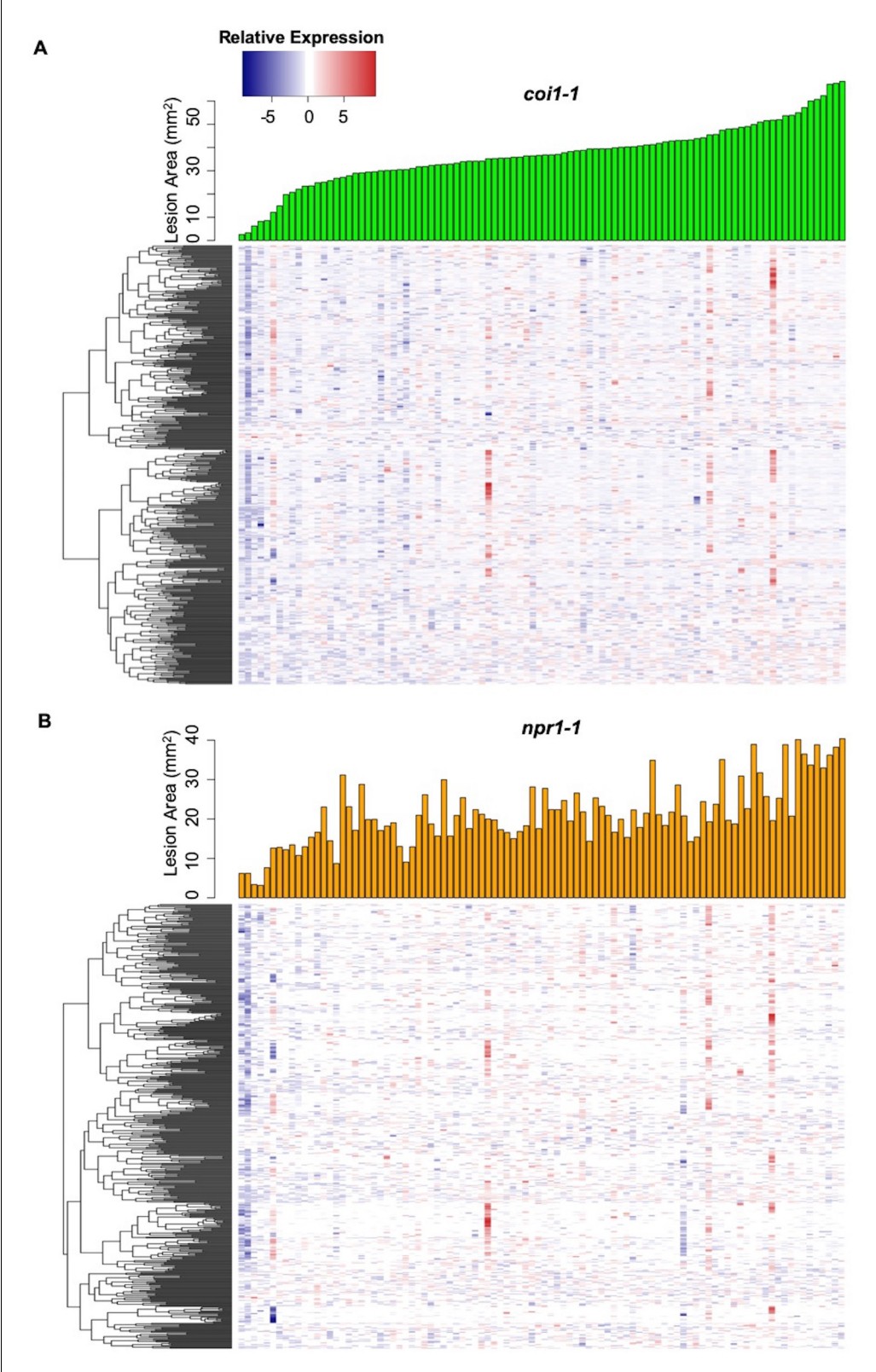

**Figure 4.** Interaction effects of host genotypes and pathogen isolates on *B. cinerea* transcriptome. Hierarchical clustering of relative expression of 500 genes from 96 *B. cinerea* isolates infection on Arabidopsis mutants *coi1-1* (**A**) or *npr1-1* (**B**) are plotted based on pairwise comparison of pathogen gene expression under Col-0. The 500 *B. cinerea* genes with highest broad-sense heritability (H$^2$) of host X pathogen were used for analysis. Lesion area induced by 96 isolates are compared under *coi1-1* (green bar plot) and *npr1-1* (orange bar plot).
*Figure 4 continued on next page*

*Figure 4 continued*

DOI: https://doi.org/10.7554/eLife.44279.013

genotypes (*Figure 7—figure supplement 4D and G*). The above observation indicates host immunity influences the *B. cinerea* transcriptional response of *B. cinerea* and suggests that *B. cinerea* isolates have varied abilities to tailor virulence strategy in response to host immunity.

## Cross-kingdom Co-transcriptomic networks revealed direct Gene-for-gene interaction

To test the interaction between individual genes from two organisms, we generated Arabidopsis-*B. cinerea* GCNs using co-transcriptome data under each host genotype. We calculated Spearman's rank correlation coefficients among 23,898 Arabidopsis transcripts and 9,284 *B. cinerea* transcripts. This approach identified three cross-kingdom GCNs (CKGCNs) under Arabidopsis WT and JA- or SA-compromised two mutants (*Figure 8*, *Supplementary file 5*, and *Figure 8—source data 1*). Under Arabidopsis WT, a total of 54 hub genes were identified, half from *B. cinerea* and half from Arabidopsis. Furthermore, CKGCNs contain a majority of genes in the BOT GCN and a small proportion of genes in the vesicle/virulence GCN (*Figure 8—figure supplement 1C*). For plants, CKGCNs contain a majority of genes from Arabidopsis Defense/camalexin GCN (*Figure 8—figure supplement 1B*). These CKGCNs also contain genes associated with extensive host defense responses, that is genes encoding membrane-localized leucine-rich repeat receptor kinases (LRR-RKs), stress signal sensing and transduction, tryptophan-derived phytoalexin production, regulation of cell death, cell wall integrity, nutrition transporters, etc. (*Figure 8—source data 1*). The topological structure and gene content of the CKGCNs shifted across the three Arabidopsis genotypes (*Figure 8*). These changes illustrate how the host genotype can influence the intercommunication in the host-pathogen interaction.

## A dual interaction network reveals fungal virulence components targeting host immunity

To begin assessing how two species influence each other's gene expression during infection, we constructed a co-transcriptome network using both the host- and pathogen-derived GCNs (*Figure 9* and *Figure 9—figure supplement 1*). We converted the ten *B. cinerea* GCNs and the four Arabidopsis GCNs into eigengene vectors that capture the variation of the general expression of all genes within a GCN into a single value (*Zhang et al., 2017*). The Arabidopsis GCNs were defined in response to this same transcriptome but by using solely the host transcripts. Of these four Arabidopsis GCNs, one is largely comprised of genes in Defense/camalexin signaling, two are linked to different aspects of photosynthesis and the fourth is largely comprised of host genes in cell division. We calculated Spearman's rank coefficients among each GCN eigengene pairs without regard for the species. In this dual transcriptome network, the Arabidopsis/*B. cinerea* GCN eigengenes are displayed as nodes and positive/negative correlations between the GCNs as edges (*Figure 9* and *Figure 9—figure supplement 1*). Of the host-derived GCNs, the Arabidopsis Defense/camalexin and Photosystem I (PSI) GCNs have a higher degree of centrality than do the Cell Division or Plastid GCNs across all three host genotypes, suggesting that they have the most interactions with *B. cinerea* GCNs. In contrast, the fungal GCNs' centrality was more dependent on the host genotype. In WT Col-0, the highest degrees were associated with the exocytosis regulation, BOT, and IPP, whereas they were more peripheral or even not present in the co-transcriptome network in the *npr1-1* or *coi1-1* host genotypes. Interestingly, in the WT Col-0 host fungal GCNs (Copper transport, Exocytosis regulation, BOT and IPP biosynthesis) that were positively correlated with the host Defense/camalexin GCN showed negative correlations with PSI eigengene. However, the host genotype can change these GCN relationships. In the *npr1-1* host, the host Defense/camalexin and PSI GCNs shift to a positive correlation. This may reflect a shift in how the *B. cinerea* BOT GCN has a positive correlation with the Defense/camalexin GCN in the Col-0 host but a negative correlation in the *npr1-1* host genotype. This suggests that there are dynamics in the host-pathogen co-transcriptome that can be interrogated to potentially identify causational relationships.

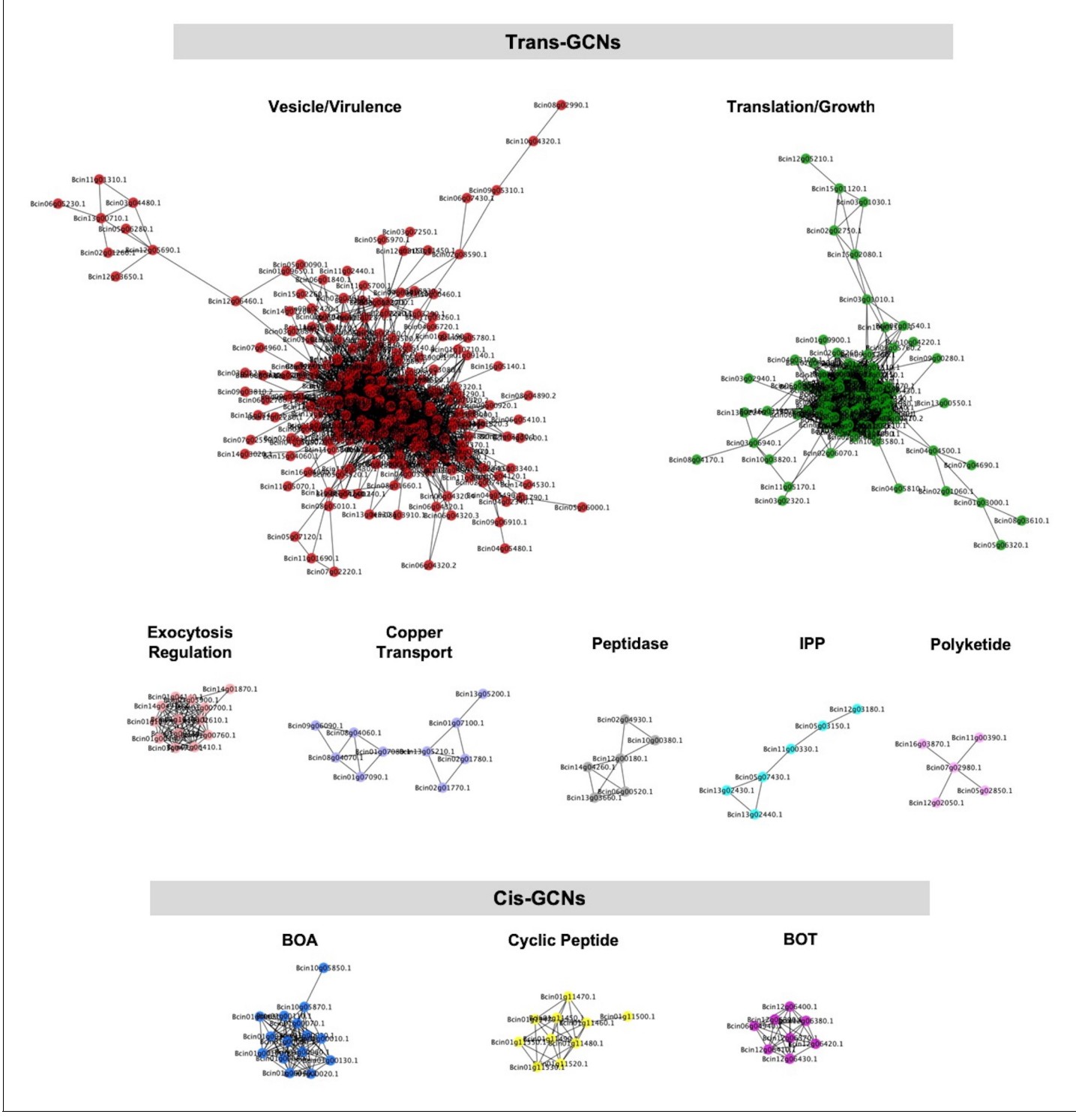

**Figure 5.** Gene co-expression networks identified from *B. cinerea* transcriptomic responses to Arabidopsis wild-type Col-0 immunity. Ten gene co-expression networks (GCNs) with more than five nodes were identified from 96 *B. cinerea* isolates infecting on Arabidopsis wild-type Col-0. The similarity matrix is computed using Spearman's rank correlation coefficient. Nodes with different colors represent *B. cinerea* genes condensed in GCNs with different biological functions. Edges represent the Spearman's rank correlation coefficients between gene pairs. Trans- and cis-GCNs means GCNs are regulated by trans- and cis-regulatory elements, respectively. GCNs were named after their biological functions, which were determined by hub and bottleneck genes within each network. GCNs are: vesicle/virulence (red), translation/growth (green), exocytosis regulation (pink), cyclic peptide (yellow), peptidase (gray), isopentenyl pyrophosphate (IPP, turquoise), polyketide (violet), botcinic acid (BOA, blue), copper transport (slate blue), botrydial (BOT, purple).

*Figure 5 continued on next page*

*Figure 5 continued*

DOI: https://doi.org/10.7554/eLife.44279.014

The following source data and figure supplement are available for figure 5:

**Source data 1.** Gene list of *B. cinerea* gene co-expression networks.

DOI: https://doi.org/10.7554/eLife.44279.016

**Figure supplement 1.** Genomic location of *B. cinerea* gene co-expression networks.

DOI: https://doi.org/10.7554/eLife.44279.015

To test if these connections were dependent upon the host immunity, we used the eigengene values derived from fungal GCNs to conduct mixed linear modelling of how they were linked to variation in the host genotype and/or host GCNs (*Supplementary file 3* and *4*). Some *B. cinerea* GCNs (Vesicle/virulence and TSL/growth, etc.) were more affected by variation in the host genotypes while others had less host dependency on their expression (BOT, Copper transport, etc.). Collectively, pathogen virulence and host immunity GCNs showed complex connections within dual interaction network identified from co-transcriptome data, suggesting functional relationships between host defense and pathogen virulence mechanisms for future experimentation.

## Germinations influence on the Co-Transcriptome

One potential complicating factor that may influence the co-transcriptome is variation in germination of the spores between *B. cinerea* strains. The lack of universal genomic patterns for host x pathogen interactions in the co-transcriptome argues that germination is not causing global effects on the co-transcriptome (*Figures 3* and *4*). To begin examining how variation in *B. cinerea* spore germination may influence the co-transcriptome and our identified link to virulence, we investigated the germination of 19 isolates. This showed that there was some variation in germination with all but a few isolates germinating within the 6–7 hr' time frame at room temperature (*Figure 9—figure supplement 2*). To extend this to an *in planta* analysis, we utilized an existing microarray study on *B. cinerea* germination to develop an eigengene that estimates the relative germination between the strains using the *in planta* transcriptomic data (*Leroch et al., 2013*). We then used this *in planta* estimation of germination to test if our previously identified co-transcriptome to virulence links were altered by controlling for germination. Using linear models, we ran the same test whereby the major *B. cinerea* GCNs were tested for a link to virulence although this time, we included the germination eigengene as a co-variate. This analysis showed that the *in planta* estimation of germination significantly associated with virulence. Critically, even with germination taken into account, all the *B. cinerea* networks remained significantly associated to lesion area. Some GCNs link, that is BOT and BOA, were largely unaffected by the germination estimates (*Supplementary file 6*), showing that some aspects of virulence are independent of spore germination. In contrast other GCNs like the vesicle linked GCN had their link to virulence decreased but not abolished by including the germination co-variate. Thus, while spore germination plays a role in our measurement of the plant-pathogen interaction, it is only one of multiple factors influencing the co-transcriptome and is not imparting a dominant global influence on the observed patterns.

## Discussion

In recent decades, improvements in the understanding of the molecular basis of plant-pathogen dynamics have facilitated breeding strategies for disease resistance in a variety of crop species. However, breeding for disease resistance remains difficult for crops susceptible to pathogens that harbor diverse polygenic virulence strategies targeting multiple layers and components of the plant innate immune system (*Corwin and Kliebenstein, 2017*). In this study, a co-transcriptomic approach was used to investigat the transcriptome profiles of both fungal pathogen *B. cinerea* and plant host Arabidopsis at an early infection stage. The results showed that the transcriptional virulence strategy employed by *B. cinerea* is dependent both on fungal genotype and the functional response of the host plant's immune system. A set of *B. cinerea* transcripts were identified with earlier expression associated with later lesion development. Furthermore, ten pathogen GCNs were found responsible for mediating virulence in *B. cinerea*, including a potential specialized metabolic pathway of cyclic

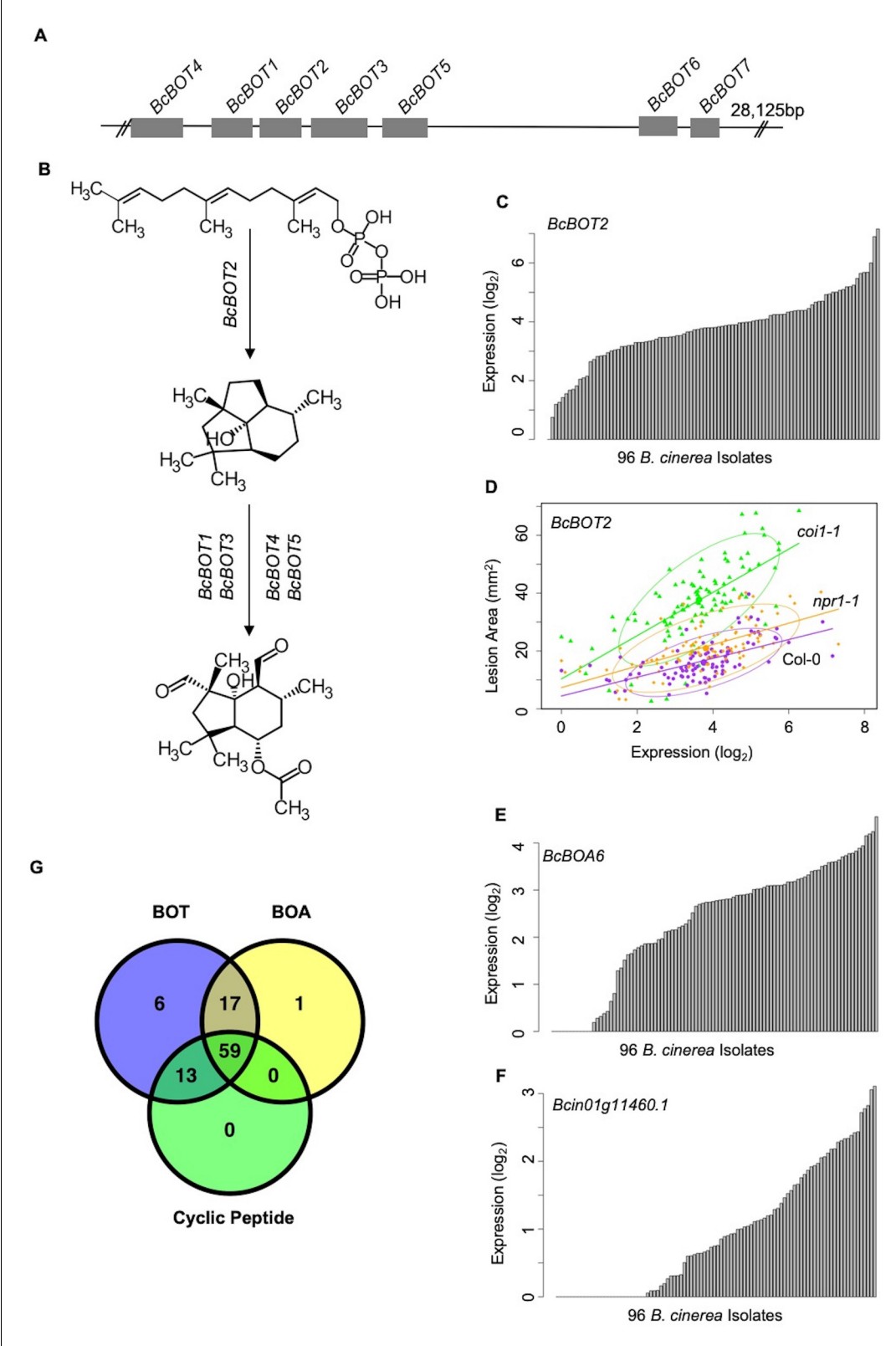

**Figure 6.** Variation of transcripts accumulation for secondary metabolites production across diverse *B. cinerea* isolates. Expression profiles of genes responsible for botrydial, botcinic acid, cyclic peptide production across 96 isolates under Arabidopsis wild-type Col-0 are shown. (A) Schematic shows the genomic locus of seven botrydial (BOT) biosynthesis genes clustered together. Exons are represented by gray boxes. Introns and intergenic regions are represented by the grey line. Seven BOT genes are: *BcBOT1*, *BcBOT3* and *BcBOT4,* encoding a cytochrome P450 monooxygenase, respectively;
*Figure 6 continued on next page*

*Figure 6 continued*

*BcBOT2* encoding a sesquiterpene cyclase; *BcBOT5* encoding an acetyl transferase; *BcBOT6* encoding a Zn(II)2Cys6 transcription factor, *BcBOT7* encoding a dehydrogenase reductase. (B) BOT biosynthesis pathway in *B. cinerea*. (C) Bar plots compare expression variation of *BcBOT2* across 96 *B. cinerea* isolates in responding to Arabidopsis wild-type Col-0 immunity. The model-corrected means (log$_2$) of transcripts were used for plotting. (D) Scatter plot illustrates the positive correlations between lesion area and accumulation of *BcBOT2* transcript across the 96 isolates in response to varied Arabidopsis immunities. Model-corrected lesion area means were estimated for three Arabidopsis genotypes at 72 hr post-infection with 96 *B. cinerea* isolates. The three Arabidopsis genotypes are labeled next to the confidence ellipse curves: wild-type Col-0 (purple dot), jasmonate insensitive mutant *coi1-1* (green triangle), and salicylic acid mutant *npr1-1* (orange diamond). The 90% confidence ellipse intervals are plotted for each Arabidopsis genotype for reference. Linear regression lines: Col-0: y = 3.2532 x+4.4323, p=1.008e-10, Adjusted R$^2$ = 3.3537; *coi1-1*: y = 7.4802 x+10.3289, p=7.895e-15, adjusted R$^2$ = 0.4700; *npr1-1*: y = 3.7086 x+7.3487, p=2.425e-11, adjusted R$^2$ = 0.3726. (E) and (F) Bar plots compare expression variation of *BcBOA6* in botcinic acid (BOA) pathway and *Bcin01g11460.* in cyclic peptide pathway across 96 *B. cinerea* isolates in response to Arabidopsis wild-type Col-0 immunity. (G) Venn diagram illustrates the number of *B. cinerea* isolates with the ability to induce BOT, BOA, and cyclic peptide.
DOI: https://doi.org/10.7554/eLife.44279.017

peptide virulence factor. Co-transcriptome networks constructed using both plant and pathogen transcriptomic data revealed known and novel fungal virulence components coordinated expressed with plant host GCNs during infection.

There are some potential limitations on the utility of cross-species GCNs. Predominantly, they are a correlational approach where links are made between host and pathogen transcriptional changes. While this leads to the development of new hypothesis, it will equally require future validation efforts to assess if these are direct or indirect relationships. Additionally, the cross-species GCN approach as implemented in this work does not distinguish between host/pathogen cells that are directly interacting versus those that are having long-distance responses. An important future avenue will be to integrate cell-specific RNA sequencing approaches to better delineate what are the responses within host/pathogen cells that are directly interacting versus the long-distance responses. This would greatly increase the power of elucidating direct versus indirect effects in this system.

## Secondary metabolites may mediate plant and fungus transcriptomic interactions during infection

Necrotrophic pathogen *B. cinerea* has evolved an arsenal of virulence strategies to establish colonization and enhance infection within the plant host, including production of secondary metabolites. The co-transcriptome approach shows that the expression of fungal specialized pathways early in infection correlates with later lesion development (*Supplementary file 3*). Three secondary metabolite GCNs are clustered within the fungal genome and two of them identified with pathway-specific transcription factors (*Figure 6*, *Figure 5*, *Figure 5—figure supplement 1*, and *Figure 5—source data 1*). Further, the expression of these pathways displayed a large range of phenotypic variation across the isolates (*Figure 6G* and *Figure 2—source data 1*). However, the topology and memberships of GCNs for the three pathways are largely insensitive to variation in host immunity. Robustness to host immunity suggests that these GCNs are somehow insulated from the host's immune response, possibly to protect toxin production from a host counter-attack. The co-transcriptome approach showed the ability to identify known and novel secondary metabolic pathways that mediate plant host and fungal pathogen interaction.

Importantly, the dual interaction networks provide hypothesis for pathogen-GCNs responsible for fungal secondary metabolites production link to specific plant host-GCNs (*Figure 9* and *Figure 9—figure supplement 1*). Specifically, the co-transcriptome approach revealed that *B. cinerea* GCNs responsible for secondary metabolite production are associated with both plant immune responses and primary plant metabolism (*Figure 9*, *Figure 9—figure supplement 1*, *Supplementary file 3* and *4*). For example, in the WT Col-0 host genotype the BOT GCN shows a strong positive correlation with the Arabidopsis Defense/camalexin GCN, suggesting that BOT production may directly induce the host's defense system. Concurrently, the BOT GCN is negatively linked to the plant's PSI GCN, suggesting that BOT may repress the plant's photosynthetic potential. Critically, this relationship changes in the *npr1-1* host genotype with the BOT GCN now having a negative correlation to the Arabidopsis Defense/camalexin GCN. Further work is needed to test if these host/pathogen GCN interactions are causal and how the SA pathway in the host may influence these interactions. Collectively, these results strongly implicate the ability of secondary metabolites biosynthesis to

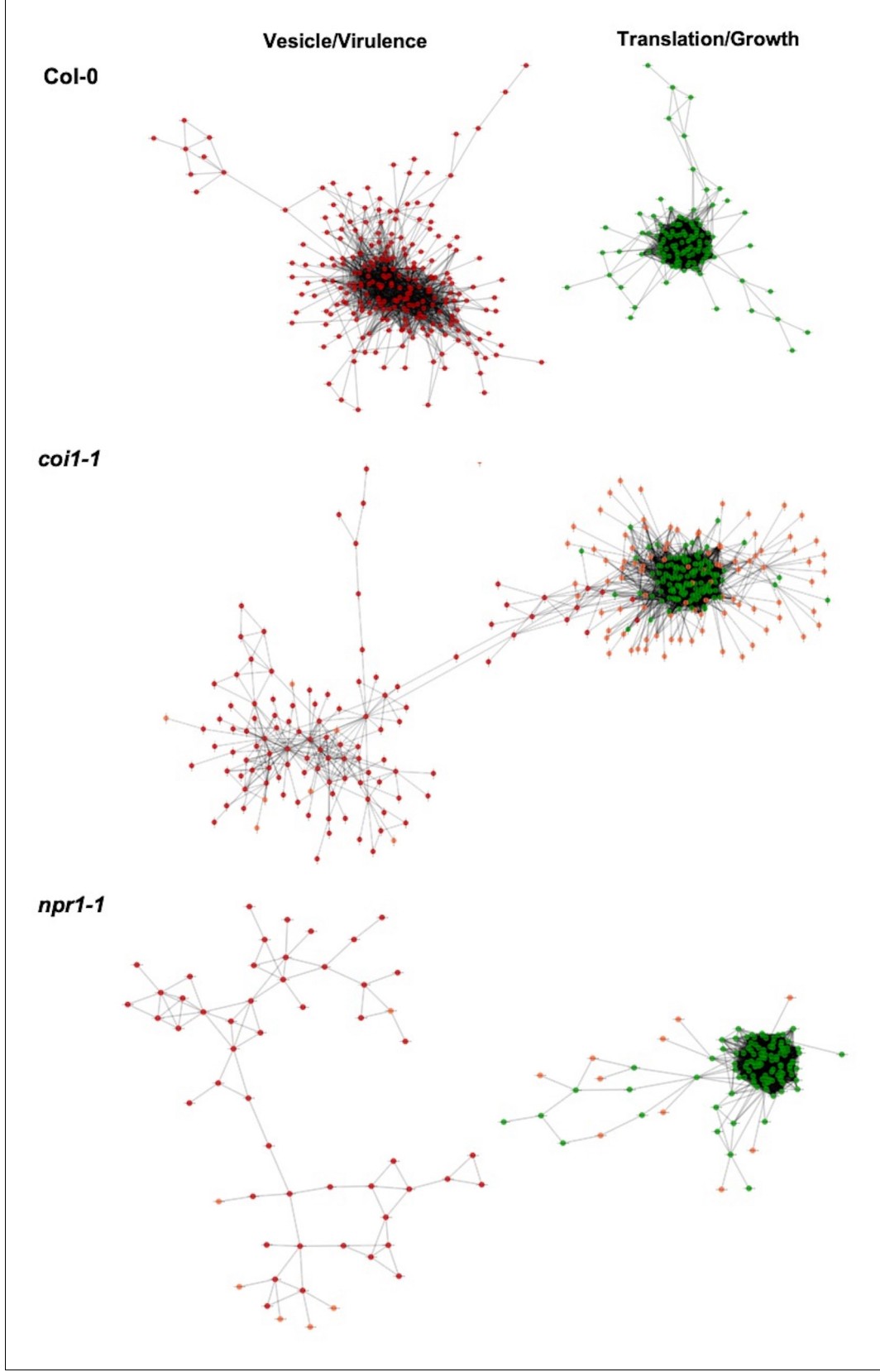

**Figure 7.** Comparison of plasticity of *B. cinerea* gene co-expression network under vaired host immunity. *B. cinerea* gene co-expression networks (GCNs) of vesicle/virulence (red) and translation/growth (green) identified

*Figure 7 continued on next page*

*Figure 7 continued*

under three Arabidopsis genotypes are compared. Three Arabidopsis genotypes are wild-type Col-0, jasmonate insensitive mutant *coi1-1*, and salicylic acid mutant *npr1-1*. Nodes marked with red and green colors represent *B. cinerea* genes condensed in GCNs with different biological functions. The same node condensed in GCNs across three Arabidopsis genotypes was marked with same color. Nodes specificaly condensed in GCNs under two mutants *coi1-1* and *npr1-1* background are marked with orange color. Edges represent the Spearman's rank correlation coefficients between gene pairs.

DOI: https://doi.org/10.7554/eLife.44279.018

The following figure supplements are available for figure 7:

**Figure supplement 1.** Gene co-expression networks identified from *B. cinerea* transcriptomic responses to Arabidopsis wild-type Col-0 immunity.

DOI: https://doi.org/10.7554/eLife.44279.019

**Figure supplement 2.** Gene co-expression networks identified from *B. cinerea* transcriptomic responses to Arabidopsis jasmonate-compromised immunity.

DOI: https://doi.org/10.7554/eLife.44279.020

**Figure supplement 3.** Gene co-expression networks identified from *B. cinerea* transcriptomic responses to Arabidopsis salicylic acid-compromised immunity.

DOI: https://doi.org/10.7554/eLife.44279.021

**Figure supplement 4.** Plasticity in expression profiles of genes identified by *B. cinerea* gene co-expression networks under varied Arabidopsis immunities.

DOI: https://doi.org/10.7554/eLife.44279.022

mediate the interactions between pathogen virulence and plant host immunity at the transcriptomic level. The co-transcriptome approach showed the potential to enable us to form new hypotheses about how this linkage may occur.

## Fungal virulence components correlated with plant immune response

In addition to secondary metabolite biosynthesis, the co-transcriptome identified a number of key virulence mechanisms that could be mapped to the two species interaction. One key GCN is enriched for genes involved in exocytosis associated regulation (*Figure 5*-Exocytosis regulation and *Figure 5—source data 1*). The exocytosis complex is responsible for delivery of secondary metabolites and proteins to the extra-cellular space and plasma membrane in fungi (*Colombo et al., 2014*; *Rodrigues et al., 2015*). Additionally, we found many *B. cinerea* genes associated with secretory vesicles within the membrane/vesicle virulence GCN that likely serve a similar function during infection (*Figure 5*-Vesicle/virulence and *Figure 5—source data 1*). These GCNs also provide support for the role of exocytosis-based spatial segregation of different materials during fungal hyphae growth *in planta* (*Samuel et al., 2015*). The dual interaction network suggests that the exocytosis regulation and membrane/vesicle virulence GCNs are differentially linked to the Arabidopsis Defense/camalexin GCN, indicating varied connections between fungal secretory pathways and plant immune responses (*Figure 9* and *Supplementary file 3* and *4*). Another conserved GCN in the *B. cinerea* species is associated with copper uptake and transport (*Figure 5*-Copper transport, *Figure 7—figure supplements 1*, *2* and *3*, and *Figure 5—source data 1*). Although copper is essential for *B. cinerea* penetration and redox status regulation within plant tissues, further work is required to decipher the precise molecular mechanism involved in acquisition and detoxification of copper. Thus, the co-transcriptome approach can identify both known and unknown mechanisms and links within the host-pathogen interaction.

## Fungal virulence transcriptomic responses are partly shaped by host immunity

It is largely unknown how plant host immunity contributes to the transcriptomic behavior of the fungus during infection. Even less is known about the role of genetic variation in the pathogen in responding to, or coping with, the inputs coming from the host immune system. In the current study, we found that the host immune system's effect on pathogen transcripts and GCNs was largely via an interaction with the pathogen genotypes (*Figure 2*, *Figure 7*, *Figure 7—figure supplement 4*, and *Figure 2—source data 4*). For example, fungal GCNs associated with membrane/vesicle virulence

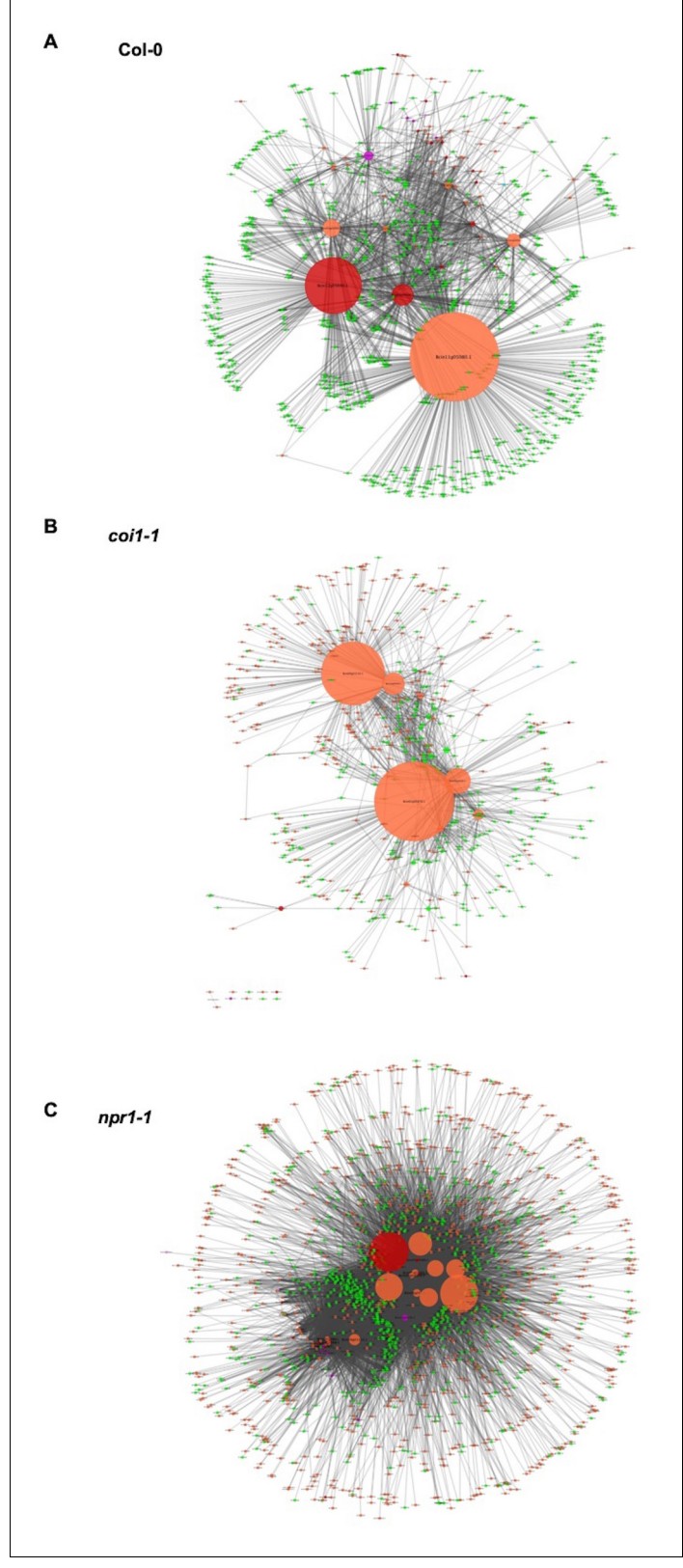

**Figure 8.** Cross-kingdom Arabidopsis-*B. cinerea* gene co-expression networks. Networks showing the co-expression connectivity between Arabidopsis and *B. cinerea* genes within three Arabidopsis genotypes are shown. (**A**) shows connectivity within Arabidopsis wild-type Col-0, (**B**) shows connectivity within the Arabidopsis jasmonate insensitive mutant *coi1-1*, and (**C**) shows connectivity within the Arabidopsis salicylic acid insensitive mutant *npr1-1*. *Figure 8 continued on next page*

*Figure 8 continued*

Within each connectivity plot, orange and green nodes show transcripts from *B. cinerea* and Arabidopsis, respectively. Nodes with red and violet colors represent the *B. cinerea* transcripts that were found to be members of the *B. cinerea* membrane/vesicle virulence network and BOT network, respectively. Node size shows the number of interactions with a specific gene. The connectivity between the nodes was derived using Spearman's rank correlation analysis.

DOI: https://doi.org/10.7554/eLife.44279.023

The following source data and figure supplement are available for figure 8:

**Source data 1.** Gene list of cross-kingdom Arabidopsis-*B. cinerea* gene co-expression networks.

DOI: https://doi.org/10.7554/eLife.44279.025

**Figure supplement 1.** Associations between gene co-expression networks identified from co- and single-transcriptome.

DOI: https://doi.org/10.7554/eLife.44279.024

and fungal growth shifted drastically between the WT and *coi1-1* or *npr1-1* Arabidopsis genotypes (*Figure 7*). In addition, some GCNs only appeared in specific backgrounds. For example, those linked to siderophores and a polyketide production were only identified during infection of the JA-compromised Arabidopsis mutant (*Figure 7—figure supplement 4J and L*). However, other fungal GCNs, like those involved in secondary metabolism, were largely insensitive to variation in the host immunity (*Figure 7—figure supplements 1*, *2* and *3*, *Supplementary file 1*, and *Figure 5—source data 1*). Critically, the gene membership of these GCNs is largely stable across the collection of pathogen isolates, even while their expression level across the *B. cinerea* isolates is highly polymorphic (*Figure 5*-source data 1and *Figure 7—figure supplement 4*). This suggests that natural variation in the host immunity and pathogen shapes how the co-transcriptome responds to host's immune system. Further, the natural variation in the pathogen may be focused around these functional GCNs.

## Plant disease development can be predicted by early transcriptome data

Plant disease development is an abstract phenomenon that is the result of a wide set of spatiotemporal biological processes encoded by two interplaying species under a specific environment. In current study, we used late stage lesion area as a quantitative indicator of *B. cinerea* virulence. We have previously shown that early Arabidopsis transcriptomic response could be linked to later lesion development (*Zhang et al., 2017*). Here, our findings suggest that the late-stage disease development of a *B. cinerea* infection is determined during the first few hours of infection by the interaction of plant immune and fungal virulence responses. It was possible to create a link between early transcripts' accumulation and late disease development using solely the *B. cinerea* transcriptome (*Figure 1* and *Figure 3—source data 1*). This could be done using either individual pathogen genes, GCNs, or more simply the total fraction of transcripts from the pathogen. As the transcriptomic data were from plant leaf tissue only 16HPI, there is not a significant amount of pathogen biomass and this is more likely an indicator of transcriptional activity in the pathogen during infection. It is possible to develop these methods as possible biomarkers for likely fungal pathogen caused disease progression.

## Conclusion

The co-transcriptome analysis of a *B. cinerea* population infection on Arabidopsis identified a number of *B. cinerea* GCNs that contained a variety of virulence-associated gene modules with different biological functions. The characterization of these GCNs simultaneously identified mechanisms known to enhance *B. cinerea* virulence and implicated several novel mechanisms not previously described in the Arabidopsis-*B. cinerea* pathosystem. In addition, the plant-fungus co-transcriptome network revealed the potential interaction between fungal pathogen- and plant host-GCNs. Construction of GCNs within single species, CKGCNs and dual networks shed lights on the biological mechanisms driving quantitative pathogen virulence in *B. cinerea* and their potential targets in the plant innate immune system.

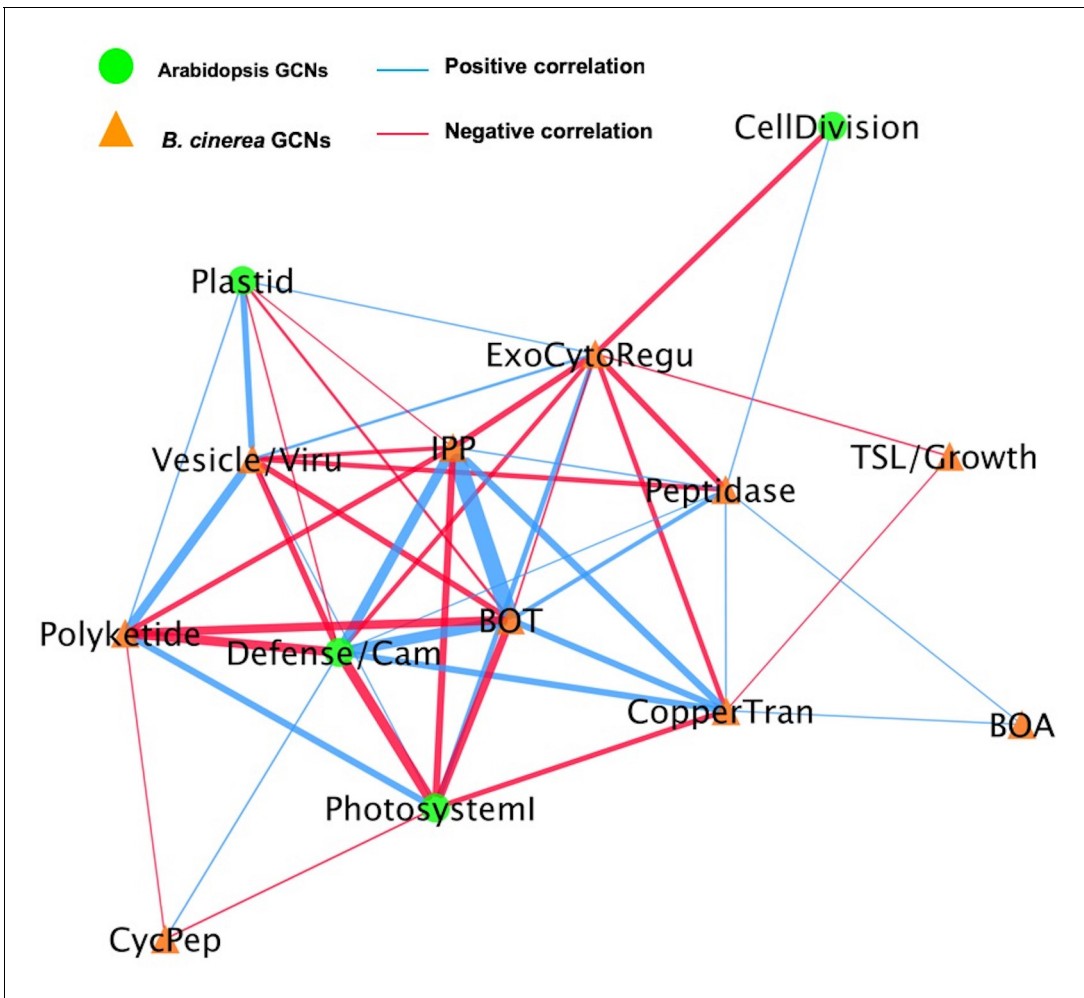

**Figure 9.** A dual interaction network reveals links between Arabidopsis immunity and *B. cinerea* virulence. A dual interaction network was constructed using gene co-expression networks (GCNs) from Arabidopsis and *B. cinerea* co-transcriptome. The first eigenvectors were derived from individual GCNs and used as input to calculate Spearman's rank correlation coefficiency between GCN pairs. Green dots and orange triangles represent Arabidopsis immune- and *B. cinerea* virulence-GCNs, respectively. Blue and red lines (edges) represent the positive and negative Spearman's rank correlation coefficients between GCN pairs, respectively. The thickness of line signifies the correlational strength.

DOI: https://doi.org/10.7554/eLife.44279.026

The following figure supplements are available for figure 9:

**Figure supplement 1.** Dual networks reveal links between Arabidopsis immunity and *B. cinerea* virulence under *coi1-1* and *npr1-1*.

DOI: https://doi.org/10.7554/eLife.44279.027

**Figure supplement 2.** Variation of *B.*

DOI: https://doi.org/10.7554/eLife.44279.028

## Materials and methods

### Collection and maintenance *B. cinerea* isolates

A collection of 96 *B. cinerea* isolates were selected in this study based on their phenotypic and genotypic diversity (*Denby et al., 2004*; *Rowe and Kliebenstein, 2007*; *Corwin et al., 2016a*; *Zhang et al., 2016*; *Zhang et al., 2017*). This *B. cinerea* collection was sampled from a large variety of different host origins and contained a set of international isolates obtained from labs across the world, including the well-studied B05.10 isolate. A majority of isolates are natural isolates that isolated from California and can infect a wide range of crops. Isolates are maintained in −80°C freezer stocks as spores in 20% glycerol and were grown on fresh potato dextrose agar (PDA) 10 days prior to infection.

## Plant materials and growth conditions

The Arabidopsis accession Columbia-0 (Col-0) was the wildtype background of all Arabidopsis mutants used in this study. The three Arabidopsis genotypes used in this study included the WT and two well-characterized immunodeficient mutants, *coi1-1* and *npr1-1*, that abolish the major JA- or SA-defense perception pathways, respectively (*Cao et al., 1997*; *Xie et al., 1998*; *Xu, 2002*; *Pieterse and Van Loon, 2004*). All plants were grown as described previously (*Zhang et al., 2017*). Two independent randomized complete block-designed experiments were conducted and a total of 90 plants per genotype were grown in 30 flats for each experiment. Approximately 5 to 6 fully developed leaves were harvested from the five-week old plants and placed on 1% phytoagar in large plastic flats prior to *B. cinerea* infection.

## Inoculation and sampling

We infected all 96 isolates onto each of the three Arabidopsis genotypes in a random design with 6-fold replication across the two independent experiments. A total of twelve infected leaves per isolate/genotype pair were generated. For inoculation, all *B. cinerea* isolates were cultured and inoculated on three Arabidopsis genotypes as described previously (*Denby et al., 2004*; *Corwin et al., 2016b*; *Zhang et al., 2017*). Briefly, frozen glycerol stocks of isolate spores were first used for inoculation on a few slices of canned peaches in petri plates. Spores were collected from one-week-old sporulating peach slices. The spore solution was filterred and the spore pellet was re-suspended in sterilized 0.5x organic grape juice (Santa Cruz Organics, Pescadero, CA). Spore concentrations were determined using a hemacytometer and suspensions were diluted to 10spores/µL. Detached leaf assays were used for a high-throughput analysis of *B. cinerea* infection, which has been shown to be consistent with whole plant assay (*Govrin and Levine, 2000*; *Mengiste et al., 2003*; *Denby et al., 2004*; *Sharma et al., 2005*; *Windram et al., 2012*). Five-week old leaves were inoculated with 4 µL of the spore solution. The infected leaf tissues were incubated on 1% phytoagar flats with a humidity dome at room temperature. The inoculation was conducted in a randomized complete block design across the six planting blocks. All inoculations were conducted within one hour of dawn and the light period of the leaves was maintained. Two blocks were harvest at 16HPI for RNA-Seq analysis. The remaining four blocks were incubated at room temperature until 72HPI when they were digitally imaged for lesion size and harvested for chemical analysis as described previously (*Zhang et al., 2017*).

## RNA-Seq library preparation, Sequencing, Mapping and Statistical Analysis

Two *B. cinerea* infected leaf tissues of the six blocks were sampled at 16HPI for transcriptome analysis, which resulted in a total of 1,052 mRNA libraries for Illumina HiSeq sequencing. RNA-Seq libraries were prepared according to a previous method (*Kumar et al., 2012*) with minor modifications (*Zhang et al., 2017*). Briefly, infected leaves were immediately frozen in liquid nitrogen and stored at −80℃ until processing. RNA extraction was conducted by re-freezing samples in liquid nitrogen and homogenizing by rapid agitation in a bead beater followed by direct mRNA isolation using the Dynabeads oligo-dT kit. First and second strand cDNA was produced from the mRNA using an Invitrogen Superscript III kit. The resulting cDNA was fragmented, end-repaired, A-tailed and barcoded as previously described. Adapter-ligated fragments were enriched by PCR and size-selected for a mean of 300 base pair (bp) prior to sequencing. Barcoded libraries were pooled in batches of 96 and submitted for a single-end, 50 bp sequencing on a single lane per pool using the Illumina HiSeq 2500 platform at the UC Davis Genome Center (DNA Technologies Core, Davis, CA). All statistical analysis were conducted within R (*R Development Core Team, 2014*).

## Transcriptomic data analysis

Fastq files from individual HiSeq lanes were separated by adapter index into individual RNA-Seq library samples. The quality of individual libraries was estimated for overall read quality and over-represented sequences using FastQC software (Version 0.11.3, www.bioinformatics.babraham.ac.uk/projects/). We conducted downstream bioinformatic analysis, like reads mapping, normalization and nbGLM model analysis, using a custom script from the *Octopus* R package (https://github.com/WeiZhang317/octopus; *Zhang, 2018*; copy archived at https://github.com/elifesciences-

publications/octopus). The mapping of processed reads against Arabidopsis and *B. cinerea* reference genomes was conducted by Bowtie 1 (V.1.1.2, http://sourceforge.net/projects/bowtie-bio/files/bowtie/1.1.2/) using minimum phred33 quality scores (*Langmead et al., 2009*). The first 10 bp of reads was trimmed to remove low quality bases using the fastx toolkit (http://hannonlab.cshl.edu/fastx_toolkit/commandline.html). Total reads for each library were firstly mapped against the Arabidopsis TAIR10.25 cDNA reference genome. The remaining un-mapped reads were then aligned against *B. cinerea* B05.10 isolate cDNA reference genome (*Lamesch et al., 2010*; *Lamesch et al., 2012*; *Krishnakumar et al., 2015*; *Van Kan et al., 2017*) and the gene counts for both species were pulled from the resulting SAM files (*Li et al., 2009*).

For pathogen gene expression analysis, we first filtered genes with either more than 30 gene counts in one isolate or 300 gene counts across 96 isolates. We normalized *B. cinerea* gene counts data set using the trimmed mean of M-values method (TMM) from the *EdgeR* package (V3.12) (*Robinson and Smyth, 2008*; *Bullard et al., 2010*; *Robinson and Oshlack, 2010*). We then ran the following generalized linear model (GLM) with a negative binomial link function from the MASS package for all transcripts using the following equation (*Venables and Ripley, 2002*):

$$Y_{egai} = E_e + E_e(Gf_g) + E_e(Gf_g(Af_a)) + I_i + H_h + H_h*I_i$$

where the main categorical effects E, I, and H are denoted as experiment, isolate genotype, and plant host genotype, respectively. Nested effects of the growing flat (Gf) within the experimental replicates and agar flat (Af) nested within growing flat are also accounted for within the model. Model corrected means and standard errors for each transcript were determined for each isolate/plant genotype pair using the lsmeans package (*Lenth, 2016*). Raw *P*-values for F- and Chi Square-test were determined using Type II sums of squares using *car* package (*Fox and Weisberg, 2011*). *P*-values were corrected for multiple testing using a false discovery rate correction (*Yoav and Daniel, 2001*). Broad-sense heritability ($H^2$) of individual transcripts was estimated as the proportion of variance attributed to *B. cinerea* genotype, Arabidopsis genotype, or their interaction effects.

## Gene Ontology analysis

GO analysis was conducted for several *B. cinerea* gene sets that were identified with high heritability, correlated with lesion size, and condensed in network analysis. We first converted sequences of these *B. cinerea* genes into fasta files using *Biostrings* and *seqRFLP* packages in R (*Qiong and Jinlong, 2012*; *Pages et al., 2017*). The functional annotation of genes was obtained by blasting the sequences against the NCBI database using Blast2GO to obtain putative GO annotations (*Conesa et al., 2005*; *Götz et al., 2008*). The GO terms were compared to the official GO annotation from the *B. cinerea* database (http://fungi.ensembl.org/Botrytis_cinerea/Info/Index) and those obtained by Blast2GO analysis. The official gene annotations for host genes was retrieved from TAIR10.25 (https://apps.araport.org/thalemine/bag.do?subtab=upload).

## *B. cinerea* Gene Co-expression Network Construction

To obtain a representative subset of *B. cinerea* genes co-expressed under *in planta* conditions, we generated gene co-expression networks (GCNs) among genes in the *B. cinerea* transcriptome. GCNs were generated using the model-corrected means of 9,284 *B. cinerea* transcripts from individual isolate infection across three Arabidopsis genotypes. Only genes with average or medium expression greater than zero across all samples were considered. This preselection process kept 6372 genes and those with negative expression values were adjusted to set expression at zero before network construction. Spearman's rank correlation coefficients for each gene pair was calculated using the *cor* function in R. Three gene-for-gene correlation similarity matrixes were generated independently for each of the three Arabidopsis genotypes. Considering the cutoff for gene-pair correlation usually generates biases of GCN structure and the candidate gene hit, we utilized several cutoff threshold values at 0.75, 0.8, 0.85, and 0.9 to filter the gene set. Comparing the structure and content of GCNs among those GCN sets using filtered gene set as input, we selected the correlation threshold at 0.8. A total of 600, 700 and 494 *B. cinerea* candidate genes passed the criterion under Arabidopsis WT, mutants *coi1-1* and *npr1-1*, respectively. To obtain a representative subset of *B. cinerea* gene candidates across three host genotypes, we selected gene candidates that presented across the above three gene subsets. This process generated a gene set with 323 *B. cinerea*

candidate genes that were common to each of the plant genotype backgrounds and had at least 0.8 significant correlations. Using this gene set as kernel, we extended gene candidate sets under each Arabidopsis genotype. The expanded *B. cinerea* gene candidate set under individual Arabidopsis genotypes was further used as input for gene co-expression network construction.

GCNs were visualized using Cytoscape V3.2.1 (Java version:1.8.0_60) (*Shannon et al., 2003*). The nodes and edges within each network represent the *B. cinerea* genes and the Spearman's rank correlations between each gene pair. The importance of a given node within each network was determined by common network analysis indices, such as connectivity (degree) and betweenness. Nodes with higher connectivity and betweenness were considered as hub and bottleneck genes, respectively, and the biological functions of each network were determined by the GO terms of hub and bottle neck genes using Blast2GO.

## Cross-kingdom Arabidopsis-*B. cinerea* Gene Co-expression Network construction

We used model-corrected means of transcripts from three Arabidopsis host genotypes and 96 *B. cinerea* isolates to construct the cross-kingdom Arabidopsis-*B. cinerea* GCNs. Model-corrected means of 23,959 Arabidopsis transcripts and 6,372 *B. cinerea* transcripts derived from two negative binomial linked generalized linear models were served as input data sets (*Zhang et al., 2017*). Spearman's rank correlation coefficient was calculated between genes from Arabidopsis and *B. cinerea* data sets. The gene pairs with positive correlations greater than 0.74 under each Arabidopsis genotype were considered to construct cross-kingdom GCNs.

## Dual interaction network construction

To construct a cross-kingdom, dual interaction network of plant-pathogen GCNs, we performed principle component analysis on individual GCNs within each species to obtain eigengene vectors describing the expression of the entire gene network as previously described (*Zhang and Horvath, 2005*; *Langfelder and Horvath, 2008*; *Okada et al., 2016*). From these eigengene vectors, we calculated the Spearman's rank correlation coefficient between the first eigengene vectors for each network. The resulting similarity matrices were used as input to construct the interaction network and Cytoscape was used to visualize the resulting network.

## Statistical analysis of network components

All the analyses were conducted using R V3.2.1 statistical environment (R Core Team, 2014). To investigate how secondary metabolite induction in *B. cinerea* contributes to disease development, we conducted a multi-factor ANOVA on *B. cinerea* three secondary metabolic pathways upon impacts on host genotypes. The three secondary metabolic pathways included the biosynthetic pathways of two well-known secondary metabolites, BOT and BOA, and a cyclic peptide biosynthetic pathway predicted in this study. We calculated the z-scores for all transcripts involved in BOT pathway, the BOA, and the putative cyclic peptide pathway for each isolate/plant genotype pair. The multi-factor ANOVA model for lesion area was:

$$y_{\text{Lesion}} = \mu + T * A * C * G_h + \epsilon$$

where T, A, C, and $G_h$ stand for BOT, BOA, Cyclic peptide, and host genotype, respectively.

In addition, we used multi-factor ANOVA models to investigate interactions between GCNs within species for impacts upon host genotypes. The ANOVA models contain all GCNs within a species. The first eigengene vector derived from principal component analysis on each network was used in ANOVA models. The ANOVA model for individual *B. cinerea* GCNs was:

$$y_{\text{BcNeti}} = \mu + D * P * C * PSI * G_h + \epsilon$$

where D, P, C, PSI, and $G_h$ stand for Arabidopsis Defense/Camalexin GCN, Arabidopsis Plastid GCN, Arabidopsis Cell/Division GCN, Arabidopsis PSI GCN, and Host genotypes, respectively. $G_h$ stands for HostGenotype, respectively. BcNeti represents one of the ten *B. cinerea* GCNs identified in this study. The ANOVA model for individual Arabidopsis GCNs was:

$$y_{\text{AtNeti}} = \mu + \Sigma \text{BcNeti} + G_h + \epsilon$$

where $\sum$BcNeti represents the summation of each of the ten *B. cinerea* GCNs identified in this study: BcVesicle/Viru GCN, BcTSL/Growth GCN, BcBOA GCN, BcExocytoRegu GCN, BcCycPep GCN, BcCopperTran GCN, BcBOT GCN, BcPeptidase GCN, BcIPP GCN, BcPolyketide GCN, while $G_h$ stands for Host genotypes. Interactions among the terms were not tested to avoid the potential for overfitting. AtNeti stands for one of the four Arabidopsis GCNs (e.g. AtDefense/Camalexin GCN, AtPlastid GCN, AtCell/Division GCN, AtPSI GCN). All multi-factor ANOVA models were optimized by trimming to just the terms with a significant *P*-value (*P*-value < 0.05).

## Germination assay

To assess the potential for natural variation in germination time in the isolate collection, 19 *B. cinerea* isolates were investigated by germination assay. The isolates were grown on PDA. Mature spores were collected in water, filtered and resuspended in 50% grape juice, as previously described, and further diluted to 1000spores/μL. To prevent germination before the beginning of the assay, spores were continuously kept on ice or in the fridge at 4°C. During the germination assay, the spores were maintained at 21°C in 1.5 mL tubes. Every hour, the tubes were mixed by manual inversion and sampled for 25 μL that were transferred to microscope slides. The spores within the drops were let to set down shortly. Without using slide covers, the spores were observed within the drops at two locations, used as technical replicates. The spores were categorized and counted based on the picture of every microscope observations taken every hour from 2 to 11 hr. Germination was defined as the hyphae emerged out of the spore.

To assess the contribution of germination to the observed *B. cinerea* transcriptomic networks involved in lesion development, we generated germination estimates based on gene expression by extracting the first principal component of a publicly available time series microarray data including 101 gemination-associated genes (*Leroch et al., 2013*). Based on this principal component, we predicted the level of expression of germination-associated genes for the 96 isolates on the three Arabidopsis genotypes at 16HPI. Theses germination predictions for individual isolates were used in a linear ANOVA model to estimate the co-linearity of the germination eigengene vector to virulence. Using linear ANOVA models with and without this germination eigengene vector, we compared how germination influences the 10 *B. cinerea* transcriptomic networks involved in lesion area in the three host genotypes. The ANOVA models with and without germination eigengene vector were:

$$Y_{\text{Lesion}} = \mu + \text{Germination} + \Sigma\text{BcNeti} + G_h + \varepsilon$$

$$Y_{\text{Lesion}} = \mu + \Sigma\text{BcNeti} + G_h + \varepsilon$$

where Germination represents the scores of first principal components on expressions of germination associated genes from *B. cinerea* transcriptomic data in this study, $\sum$BcNeti represents the summation of each of the ten *B. cinerea* GCNs identified in this study: BcVesicle/Viru GCN, BcTSL/Growth GCN, BcBOA GCN, BcExocytoRegu GCN, BcCycPep GCN, BcCopperTran GCN, BcBOT GCN, BcPeptidase GCN, BcIPP GCN, BcPolyketide GCN, while $G_h$ stands for Host genotypes. Interactions among the terms were not tested to avoid the potential for overfitting.

## Additional information

### Competing interests

Daniel J Kliebenstein: Reviewing editor, *eLife*. The other authors declare that no competing interests exist.

### Funding

| Funder | Grant reference number | Author |
| --- | --- | --- |
| National Science Foundation | IOS 1339125 | Daniel J Kliebenstein |
| U.S. Department of Agriculture | Hatch project number CA-D-PLS-7033-H | Daniel J Kliebenstein |

| Danish National Research Foundation | DNRF99 | Daniel J Kliebenstein |
|---|---|---|
| China Scholarship Council | 20130624 | Wei Zhang |

The funders had no role in study design, data collection and interpretation, or the decision to submit the work for publication.

## Author contributions
Wei Zhang, Conceptualization, Resources, Data curation, Software, Formal analysis, Validation, Visualization, Methodology, Writing—original draft, Writing—review and editing; Jason A Corwin, Conceptualization, Resources, Data curation, Software, Formal analysis, Methodology, Writing—review and editing; Daniel Harrison Copeland, Julie Feusier, Robert Eshbaugh, Experiment conduction, Data collection; David E Cook, Writing—review and editing; Suzi Atwell, Resources, Data curation, Software; Daniel J Kliebenstein, Conceptualization, Data curation, Formal analysis, Supervision, Funding acquisition, Validation, Investigation, Methodology, Project administration, Writing—review and editing

## Author ORCIDs
Wei Zhang http://orcid.org/0000-0002-5092-643X
Daniel Harrison Copeland https://orcid.org/0000-0002-2206-9127
David E Cook http://orcid.org/0000-0002-2719-4701
Daniel J Kliebenstein https://orcid.org/0000-0001-5759-3175

## Decision letter and Author response
Decision letter https://doi.org/10.7554/eLife.44279.041
Author response https://doi.org/10.7554/eLife.44279.042

# Additional files

## Supplementary files
• Supplementary file 1. Topology traits of *B. cinerea in planta* gene co-expression networks.
DOI: https://doi.org/10.7554/eLife.44279.029

• Supplementary file 2. ANOVA table of lesion area and *B. cinerea* pathways. A mixed linear model was fitted to test lesion area and *B. cinerea* pathways responsible for botrydial (BOT), botcinic acid (BOA), and cyclic peptide (CycPep) produced under three Arabidopsis genotypes. The lesion area data used in the model were GLM corrected least square means induced by 96 *B. cinerea* isolates. Model-corrected means of transcripts from 96 *B. cinerea* isolates were z-scaled and used in ANOVA. Df is the degrees of freedom for a term within the model. SS is the Sum of Squares variation. MS is the Mean of Squared variation. F value is derived from the F statistic and *P*-value indicates the statistical significance for a given term within the model. Significance of differences are shown as p<0.001 '***', 0.01'**' and 0.05 '*'.
DOI: https://doi.org/10.7554/eLife.44279.030

• Supplementary file 3. ANOVA tables of *B. cinerea* gene co-expression networks. Mixed linear models were fitted to individual *B. cinerea* (Bc) gene co-expression networks (GCNs) and variation of host genotypes and Arabidopsis (At) GCNs. Variation was estimated among host genotypes and first eigenvectors from four individual Arabidopsis GCNs. Df is the degrees of freedom for a term within the model. SS is the Sum of Squares variation. MS is the Mean of Squared variation. F value is derived from the F statistic and *P*-value indicates the statistical significance for a given term within the model. Significance of difference are shown as p<0.001 '***', 0.01'**' and 0.05 '*'.
DOI: https://doi.org/10.7554/eLife.44279.031

• Supplementary file 4. ANOVA tables of Arabidopsis gene co-expression networks. Linear mixed models were fitted to individual Arabidopsis (At) gene co-expression networks (GCNs) and variation of host genotypes and ten *B. cinerea* (Bc) GCNs. Variation was estimated among host genotypes and first eigenvectors from individual *B. cinerea* GCNs. Df is the degrees of freedom for a term within the model. SS is the Sum of Squares variation. MS is the Mean of Squared variation. F value is

derived from the F statistic and *P*-value indicates the statistical significance for a given term within the model. Significance of difference are shown as p<0.001 '***', 0.01'**' and 0.05 '*'.
DOI: https://doi.org/10.7554/eLife.44279.032

• Supplementary file 5. Topology traits of cross-kingdom Arabidopsis-*B. cinerea* gene co-expression networks. Three Arabidopsis genotypes are wild-type Col-0, jasmonate insensitive mutant *coi1-1*, and salicylic acid insensitive mutant *npr1-1*.
DOI: https://doi.org/10.7554/eLife.44279.033

• Supplementary file 6. Analysis of potential impact of germination variation. To test if germination may influence the observed network to lesion connections, we estimated germination using the first principal component of genes linked to germination in *Leroch et al. (2013)*. We then estimated the value of this principal component in the isolates grown on the three host genotypes and conducted a linear model to compare how this eigengene links to virulence. Using a linear model with and without this germination eigengene, we compared the link of the 10 *B. cinerea* transcript networks to lesion size with and without the germination eigengene vector in the model.
DOI: https://doi.org/10.7554/eLife.44279.034

• Transparent reporting form
DOI: https://doi.org/10.7554/eLife.44279.035

## Data availability

The datasets used in this study are available in the following database: Bioproject PRJNA473829 (https://www.ncbi.nlm.nih.gov/bioproject/?term=PRJNA473829). The computer scripts used in this study are available in GitHub (https://github.com/WeiZhang317/octopus; copy archived at https://github.com/elifesciences-publications/octopus).

The following previously published datasets were used:

| Author(s) | Year | Dataset title | Dataset URL | Database and Identifier |
|---|---|---|---|---|
| Zhang W, Corwin JA, Copeland D, Feusier J, Eshbaugh R, Chen F, Atwell S, Kliebenstein DJ | 2017 | Data from: Plastic transcriptomes stabilize immunity to pathogen diversity: the jasmonic acid and salicylic acid networks within the Arabidopsis/Botrytis pathosystem | http://dx.doi.org/10.5061/dryad.7gd5q | Dryad, 10.5061/dryad.7gd5q |
| Zhang W, Corwin JA, Copeland D, Feusier J, Eshbaugh R, Chen F, Atwell S, Kliebenstein DJ | 2017 | Plastic Transcriptomes Stabilize Immunity to Pathogen Diversity | https://www.ncbi.nlm.nih.gov/bioproject/PRJNA473829 | NCBI Bioproject, PRJNA473829 |

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
