## [Decision Letter]

Thank you for submitting your article "Plant-necrotroph co-transcriptome networks illuminate a metabolic battlefield" for consideration by *eLife*. Your article has been evaluated by three peer reviewers, one of whom is a member of our Board of Reviewing Editors, and the evaluation has been overseen by Ian Baldwin as the Senior Editor. The reviewers have opted to remain anonymous.

The reviewers have discussed the reviews with one another and the Reviewing Editor has drafted this decision to help you prepare a revised submission.

Summary:

We all agree that this work is substantial and of a high quality and provides interesting insights. Co-transcriptome analysis during plant-microbe interaction is very rare in literature, and therefore this work is highly novel. Nevertheless, the reviewers raised three major issues and we discussed those together.

A first issue is the lack of demonstrated causality for newly generated hypotheses. These would strengthen the manuscript and validate the co-transcriptome analyses. After extensive discussion, we, however, concluded that this is not essential for revision and is beyond the scope of the current manuscript.

A second issue is if in planta *B. cinerea* transcriptomes were determined "solely" by *B. cinerea* germination and developmental timing of different isolates. With 96 isolates, we expect that timing of germination or development of some *B. cinerea* isolates differ. Thus, some *B. cinerea* transcriptome responses would be determined by developmental timing of isolates but not virulence mechanisms. If this were the case for all strains, authors' major conclusions are not justified. We concluded that you need to demonstrate that in planta *B. cinerea* transcriptome patterns are "not solely" determined by germination and developmental timing.

A third issue is while you provided summary statistics in the supplement, most readers will not easily understand them. For instance, you showed that host immunity affected *B. cinerea* transcriptomes in a strain-specific manner as exemplified in Figure 2A and Figure 3. While you described this for some *B. cinerea* genes, readers would for sure appreciate to know to what extent (and for which *B. cinerea* genes and biological processes) host immunity affects fungal gene expression in a strain-specific or a similar manner.

The reviewers also raised some points for improvement of the manuscript. Please consider these points when you prepare your revision. Note that these are not essential for your revision but suggestions.

Essential revisions:

1) You need to provide evidence for that in planta *B. cinerea* transcriptome patterns are not solely determined by germination and developmental timing of *B. cinerea* isolates in plants. We request you to test this experimentally. We suggest that you investigate germination and development of 10 (instead of 96) *B. cinerea* isolates in Col-0 plants, which showed differential transcriptome patterns in plants, by quantifying *B. cinerea* biomass and staining *B. cinerea*. Depending on the result, you need to modify some of your conclusions.

2) You need to provide a genome-wide analysis for host X isolate specific interactions that is more accessible to readers. One way the authors can do is to expand Figure 3 analysis to the *B. cinerea* genome wide. We suggest that the authors standardize each *B. cinerea* gene (relative expression) so that plant genotype effects can be easily seen. Then, clustering (both column and row directions) followed by visualization in a heat map can be performed using such expression table with y-axis being *B. cinerea* genes and x-axis being *B. cinerea* strains in a plant genotype. At the same time, visualizing disease scores (bar graphs?) on top of column would be also informative. Then, these analyses (maybe in part) can be shown as a main content. These analyses would allow readers to see which *B. cinerea* genes and processes are influenced by the host immune pathways in a strain specific or non-specific manner at the global scale. It might be better to use log fold change (*coi1-1* vs. Col and *npr1-1* vs. Col). In this case, analyzing *coi1-1* vs. Col and *npr1-1* vs. Col separately may generate a clearer pattern.

---

## [Author Response]

Summary:We all agree that this work is substantial and of a high quality and provides interesting insights. Co-transcriptome analysis during plant-microbe interaction is very rare in literature, and therefore this work is highly novel. Nevertheless, the reviewers raised three major issues and we discussed those together.A first issue is the lack of demonstrated causality for newly generated hypotheses. These would strengthen the manuscript and validate the co-transcriptome analyses. After extensive discussion, we, however, concluded that this is not essential for revision and is beyond the scope of the current manuscript.

We appreciate the reviewers’ perspective regarding the issues to strengthen and validate the in planta co-transcriptome analyses. We have worked to respond to all the specific requests for additional information including experimental assay and data analysis. This is detailed in the individual responses to the reviewers.

A second issue is if in planta B. cinerea transcriptomes were determined "solely" by B. cinerea germination and developmental timing of different isolates. With 96 isolates, we expect that timing of germination or development of some B. cinerea isolates differ. Thus, some B. cinerea transcriptome responses would be determined by developmental timing of isolates but not virulence mechanisms. If this were the case for all strains, authors' major conclusions are not justified. We concluded that you need to demonstrate that in planta B. cinerea transcriptome patterns are "not solely" determined by germination and developmental timing.

We have worked on both experimental and in silico assay to estimate how germination may or may not influence on the links between in planta fungal transcriptomic networks and lesion development. We investigated 19 *B. cinerea* isolates by germination assay. We further estimated variation in germination *in silico* by using an existing microarray study and the in planta transcriptomic data (citation). The results showed while spore germination plays a role in the plant-pathogen interaction, all the *B. cinerea* networks that were significant without germination remained significantly associated to lesion area when including germination. Thus, germination is only one of multiple factors influencing the interaction. We added a new section in the main text together with Figure 9—figure supplement 2 and Supplementary File 6 to show the results.

A third issue is while you provided summary statistics in the supplement, most readers will not easily understand them. For instance, you showed that host immunity affected B. cinerea transcriptomes in a strain-specific manner as exemplified in Figure 2A and Figure 3. While you described this for some B. cinerea genes, readers would for sure appreciate to know to what extent (and for which B. cinerea genes and biological processes) host immunity affects fungal gene expression in a strain-specific or a similar manner.

We agree with the reviewers that it would be informative to visualize the host X isolate interaction effects at the global scale. We have generated a clustered heat map (Figure 4) to compare the fold change of 500 *B. cinerea* genes controlled by host X isolate induced under *coi1-1* vs. Col-0 or *npr1-1* vs. Col-0. Going to a larger set of genes leads to a very unwieldy figure. We hope that this helps to generate a better view of the complexity and how each of the strains are behaving in highly individualized ways. We had tried to utilize common summary approaches like GO annotation. However, these are not as useful as we would like in organisms with relatively small research communities like *B. cinerea*. We hope that in combination with the additional co-transcriptome plots and the new genome wide host x strain figure (Figure 4) that we have helped to improve this aspect of the manuscript.

Essential revisions:1) You need to provide evidence for that in planta B. cinerea transcriptome patterns are not solely determined by germination and developmental timing of B. cinerea isolates in plants. We request you to test this experimentally. We suggest that you investigate germination and development of 10 (instead of 96) B. cinerea isolates in Col-0 plants, which showed differential transcriptome patterns in plants, by quantifying B. cinerea biomass and staining B. cinerea. Depending on the result, you need to modify some of your conclusions.

We have worked in both wet lab and in silico to estimate how germination variation may influence our observed links between in planta fungal transcriptomic networks and lesion development. Please see the germination assay in Materials and methods section and the new germination section of the manuscript together with the Figure 9—figure supplement 2 and Supplementary file 6 in the Results section. We feel that this in combination with Figure 4 and the other figures shows that germination is not a singular universal driver of the observed patterns and their link to virulence.

2) You need to provide a genome-wide analysis for host X isolate specific interactions that is more accessible to readers. One way the authors can do is to expand Figure 3 analysis to the B. cinerea genome wide. We suggest that the authors standardize each B. cinerea gene (relative expression) so that plant genotype effects can be easily seen. Then, clustering (both column and row directions) followed by visualization in a heat map can be performed using such expression table with y-axis being B. cinerea genes and x-axis being B. cinerea strains in a plant genotype. At the same time, visualizing disease scores (bar graphs?) on top of column would be also informative. Then, these analyses (maybe in part) can be shown as a main content. These analyses would allow readers to see which B. cinerea genes and processes are influenced by the host immune pathways in a strain specific or non-specific manner at the global scale. It might be better to use log fold change (coi1-1 vs. Col and npr1-1 vs. Col). In this case, analyzing coi1-1 vs. Col and npr1-1 vs Col separately may generate a clearer pattern.

We have generated a heat map (Figure 4) to visualize fold changes of 500 *B. cinerea* genes controlled by host x isolate specific effects under two Arabidopsis mutants, which was plotted based on reviewers’ suggestion. We have ordered the isolates based on their virulence on the *coi1-1*-1 genotype and shown the virulence on the same figure. We hope that it is informative and helpful for readers to understand the host x pathogen effects at a global scale.